# Fast objective coupled planar illumination microscopy

Cody J. Greer [ID] [1]* & Timothy E. Holy[1]

Among optical imaging techniques light sheet fluorescence microscopy is one of the most attractive for capturing high-speed biological dynamics unfolding in three dimensions. The technique is potentially millions of times faster than point-scanning techniques such as two-photon microscopy. However light sheet microscopes are limited by volume scanning rate and/or camera speed. We present speed-optimized Objective Coupled Planar Illumination (OCPI) microscopy, a fast light sheet technique that avoids compromising image quality or photon efficiency. Our fast scan system supports 40 Hz imaging of 700 μm-thick volumes if camera speed is sufficient. We also address the camera speed limitation by introducing Distributed Planar Imaging (DPI), a scaleable technique that parallelizes image acquisition across cameras. Finally, we demonstrate fast calcium imaging of the larval zebrafish brain and find a heartbeat-induced artifact, removable when the imaging rate exceeds 15 Hz. These advances extend the reach of fluorescence microscopy for monitoring fast processes in large volumes.

[1] Department of Neuroscience, Washington University in Saint Louis, Saint Louis, MO 63110, USA. *email: cody.greer@wustl.edu

I mprovements in calcium[1] and voltage[2,3] indicators have opened up the possibility of using fluorescence microscopy to observe interactions between neurons in the brain. Because neuronal computations occur on short timescales and are distributed over large tissue volumes, capturing these three-dimensional dynamics presents a major acquisition challenge. Motivated by this goal, many groups have developed or refined microscopy techniques that increase the scale and/or speed of fluorescence microscopy[4–24]. These fast techniques can be broadly categorized by the way that they collect pixel information to form an image: point-scanning (confocal and two-photon) methods collect one or a few[16] pixels at a time, while line-scanning and widefield methods parallelize acquisition over hundreds to millions of pixels on a camera sensor.

Fundamental physics—specifically, limits on the relaxation time of fluorophores, leading to a phenomenon known as fluorophore saturation[25]—dictates that single point-scanning imaging techniques have lesser potential speed than techniques that image many points in parallel. Traditional widefield imaging methods such as epifluorescence excel at this parallelization but pay a cost in axial resolution: they produce 2-dimensional images containing a mixture of fluorescent sources from multiple depths in the sample.

Recently several approaches have been developed to separate signals by depth–referred to as optical sectioning–while imaging many points in parallel. Light-sheet fluorescence microscopy (LSFM) is one of the most popular of these approaches. Conventional LSFM performs optical sectioning by way of specialized hardware that limits excitation light to a single plane at a time while capturing an image of that plane with a camera. Thus the illuminated region is coincident with the image plane, allowing the entire image to be captured in parallel, while also avoiding unnecessary light exposure and affording LSFM with exceptionally low phototoxicity. Other techniques, as well as some LSFM variants use a combination of specialized hardware and deconvolution software[5,12,14,15,17,24]. While such techniques are among the fastest available, they make tradeoffs in image quality, and the computationally intensive postprocessing can become impractical for large datasets. In this study we sought to develop a hardware-only LSFM that avoids these tradeoffs when imaging large volumes of tissue.

If one is only interested in imaging a single plane of tissue with LSFM, then the typical 0.1 μs to 1.0 μs voxel dwell time implies the potential for acquiring $10^6$–$10^7$ frames/s; in practice, for sustained imaging the rate of all current microscopes is limited by the frame rate of the camera. However when imaging a 3-dimensional volume another rate-limiting factor may come into play: the speed of the mechanism by which the image and illumination planes are repositioned within the sample to compose a stack of images. We will refer hereafter to this potential bottleneck as the volume scanning bottleneck to distinguish it from the camera frame rate bottleneck.

Several variants of LSFM have been developed, and they differ in the extent to which they suffer from the scanning bottleneck. Classic LSFM variants such as Selective Plane Illumination Microscopy[4] (SPIM) and OCPI microscopy[6] are limited more severely by the volume scanning bottleneck than the camera frame rate bottleneck. Some newer techniques support almost unlimited volume scanning speed, but we show that relative to OCPI and SPIM all of these designs compromise photon efficiency[8,10,15,17,22,23,26–28] and/or spatial resolution[10,15,17,19,26,27,29]. As a consequence, none of these is capable of scanning volumes hundreds of microns on a side without compromising optical quality.

In this study, we identify and analyze the factors that give rise to the scan bottleneck in OCPI microscopy and resolve them in a way that avoids making such sacrifices. Having overcome the volume scanning bottleneck, we then alleviate the camera frame rate bottleneck by debuting DPI to parallelize imaging across multiple cameras. These contributions are not specific to OCPI microscopy: other systems can exploit our mechanical scanning optimizations, and any system that images a plane onto a camera sensor can be sped up with DPI. To facilitate adoption of our specific microscope by other research groups, we provide detailed hardware schematics and open-source software. This microscope serves more than a dozen users in a centralized imaging facility, demonstrating its flexibility and robustness.

Finally we provide two imaging demonstrations using our microscope: imaging neural activity in the whole brain of a larval zebrafish at 10 Hz, as well as imaging just the forebrain at 20 Hz, in both cases with a lateral sampling rate of 0.65 μm pixel$^{-1}$. We analyze the forebrain recording and show that neuronal traces are contaminated with an artifact arising from the heartbeat of the fish. We demonstrate that this artifact introduces spurious correlations between neurons, and that a sampling rate of at least 15 Hz is required in order to remove the artifact. Expunging this artifact will be an important processing step for future large-scale studies of zebrafish brain activity.

## Results

**Factors limiting imaging rate and image quality with LSFM microscopy.** Recent work has emphasized remote focusing[30,31] methods that image voxels outside of the objective's native focal plane, avoiding mechanical translation in order to scan faster. These methods have a critical disadvantage relative to SPIM or OCPI: they trade away image quality in favor of scan speed. Some suffer from reduced photon efficiency[8,10,15,17,22,23,26–28], with a representative example[28] losing 79% of light collected by the objective lens. Some contend with aberrated images[10,15,17,19,26,27,29]. These aberrations are primarily spherical, and their severity increases with distance from the native focal plane, magnification, and numerical aperture (NA) of the system. The aberrations are not a technical limitation of existing lenses: in Supplementary Note 1, we present a proof that they follow directly from the Abbe sine condition and apply to any magnifying system that collects images away from the classical focal plane. Figure 1a shows the theoretical performance of such a remote focusing system using three common microscope objectives at various magnifications and NA values (derived in Supplementary Note 1). At NA 0.3, diffraction-limited axial resolution is limited to an area within 385 μm from the focal plane. At NA 0.5 (2.9× the photon efficiency of NA 0.3) this is reduced to 47 μm, and then falls to only 6 μm at NA 0.8. OCPI and SPIM do not suffer this tradeoff, enabling them to scale to axial spans of hundreds of microns while enjoying the resolution and efficiency of higher NAs.

This analysis suggests that approaches collecting images using only the native focal plane of the objective, as performed in SPIM and OCPI microscopy, merit re-examination. Scaleable to large samples with uniform resolution and minimal photodamage, these methods are well-suited to address a current challenge in optical neurophysiology: repeatedly imaging thousands of living neurons.

Figure 1b illustrates the basic OCPI microscope design and shows that both the imaging objective and the lightsheet optics are translated (scanned) together in the axial direction. Scanning with OCPI is accomplished with a linear actuator, usually a piezoelectric device. Due to the use of infinity-focused optics, this translation does not affect imaging quality or efficiency, allowing diffraction-limited imaging throughout the range of the actuator. An empirical measurement of the OCPI axial point spread

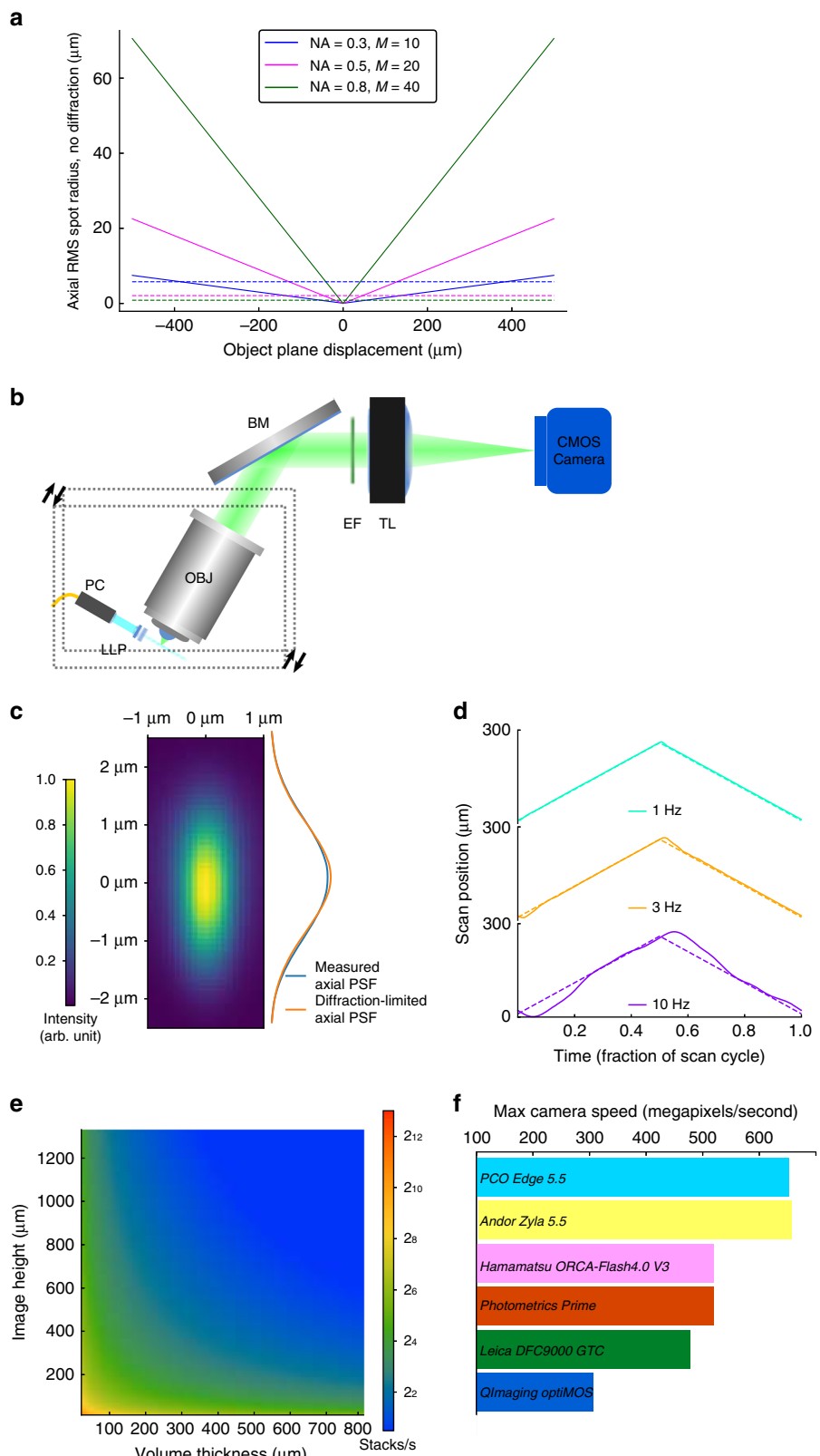

function (PSF) (Fig. 1c) confirms that the system is essentially diffraction-limited ($NA_{detect} = 0.5$ and $NA_{illum} = 0.11$; sheet thickness was roughly matched to the objective depth-of-field).

Unfortunately, volume imaging with SPIM and OCPI is usually rate-limited by the inertia of the sample or the optics, respectively. Due to this inertia the piezo positioner fails to

follow high frequency commands (Fig. 1d). This failure can have catastrophic consequences for 3D imaging at high rates. Unwanted oscillations in the scan system can result in both undersampling and oversampling in the axial dimension. Optical sections may also appear out-of-order since the axial position may not be strictly increasing/decreasing between sections.

**Fig. 1** Volume imaging advantages and rate bottlenecks of OCPI microscopy. **a** Theoretically-estimated axial spherical aberrations for a microscope employing remote focusing without aberration correction (see Supplementary Note 1 derivation). Plotted are spot sizes (in object space) for three objective lenses (Olympus UMPLFLN family). OCPI microscopy is limited only by the diffraction limit (dashed lines, Gaussian approximation, see Methods section). **b** Schematic of an OCPI microscope with minor modifications to the design described in previous work[6,7]. Optical sectioning is achieved by translating the optics for generating the light sheet together with the detection objective (boxed components). PC pigtailed collimator, LLP light sheet lens pair, OBJ objective, BM broadband mirror, EF emission filter, TL tube lens. **c** OCPI microscopes achieve diffraction-limited axial resolution throughout the scan range because scanning changes only the infinity-focused portion of the light path. Shown is an OCPI point spread function found by averaging images of fluorescent beads throughout a volume. The axial intensity distribution matches closely the diffraction limit predicted by Gaussian optical theory (compare blue and orange curves, $NA_{det} = 0.5$, $NA_{illum} = 0.11$, details in methods). **d** Inertia of the boxed components in (b) limits volume scanning rate with OCPI. Shown are triangle wave scan commands (dashed lines) and measured responses (solid lines) from the piezoelectric positioner of an unoptimized OCPI microscope. One full scan cycle is shown at each of the three scan rates. The scan waveform response is distorted at higher scan rates due to inertia and imperfect closed-loop control. **e** If volume scanning were not rate-limiting, OCPI microscopy volume imaging rate would scale with the height of each camera image, as well as the thickness of the imaged volume (shown for ×10 magnification with 5 μm spacing of optical sections, PCO.Edge 4.2 camera). We sought to optimize the scan system so that scan rate was not limiting in most of the parameter space shown. **f** Maximum pixel rates of popular 16-bit scientific CMOS cameras are shown. These maximum data rates limit all current widefield microscopy methods. No high-sensitivity camera released within the past 6 years has improved upon the rates shown, motivating us to circumvent this limitation

Particular high-frequency commands may even send the piezo system's closed-loop controller into an unstable regime that destroys the device. If not for the scanning bottleneck, volume imaging rate scales with the frame rate of the camera (this is true for all LSFM methods, not only for OCPI). The achievable volume rate is inversely proportional to the size of the volume and the density at which the volume is sampled. This scaling is illustrated for a modern scientific CMOS camera (PCO.Edge 4.2) in Fig. 1e, and other scientific camera models perform similarly (Fig. 1f). To our knowledge the fastest volume imaging demonstrated with OCPI or SPIM was 5 Hz with a 200 μm scan range[32] (an average scanning speed of 2 mm s$^{-1}$ when considering both the forward and reverse scan sweeps). Thus scanning limitations prevent realization of most of the imaging rates shown in Fig. 1e, and we sought to access this unused camera capacity.

**Fast mechanical scanning**. In order to address the scanning bottleneck we enacted five design strategies: (1) minimizing the mass of all translated components, (2) optimizing the command signals that drive the piezoelectric actuator, (3) calibrating the timing of camera exposures, (4) pulsing the illumination laser during the global exposure period of the camera, and (5) acquiring image stacks during both the forward and reverse sweeps of the scan. A photo of our mass-optimized scan assembly is shown in Fig. 2a. Rather than create the lightsheet with a second objective we used custom optics of minimal size[6], and mounting and alignment hardware was machined to minimize mass while maintaining rigidity.

Our chosen positioner was able to generate push and pull forces of up to 100 N, more than enough force to achieve the accelerations necessary to scan at frequencies up to 20 Hz. To counteract the piezoelectric phenomena of creep and hysteresis[33], we elected to use this with a closed-loop controller. However, the performance of a closed-loop piezo system is particularly sensitive to the mass of translated components, center of gravity, and translation angle relative to gravity. The controller in our system is of the proportional integral derivative (PID) variety. We requested that the vendor optimize the three tunable PID parameters for the mass of our assembly and angle of translation. Additional tuning was performed manually so that the system's response matched high-frequency scanning commands as closely as possible (see Methods section). We also verified that the response of the piezo system to a cyclic command was highly consistent across cycles (Supplementary Fig. 1), which is crucial for stable multi-stack recordings.

PID control is quite sensitive to large accelerations in the command signal such as those at the extrema of a triangle wave,

causing the system to exhibit unfavorable higher frequency oscillations (Fig. 1d). We addressed this issue by utilizing a lowpass filtered triangle wave command, where the filter had a cutoff of 3.25× the command frequency (see Methods section). We also performed a brief iterative optimization of the amplitude and offset of the command signal to achieve the desired scan range as measured by the sensor (see Methods section). Thus by tuning both the PID and the command waveform we were able to drive the piezo smoothly through a range of up to 700 μm at frequencies up to 20 Hz (Fig. 2b).

**Sensor-guided exposure timing**. For later analysis steps such as image registration, it is desirable for the slices of an image stack to be equally spaced in the axial direction. Because of the non-uniform piezo speed, collecting camera frames with a fixed frame rate would result in image stacks that do not meet this requirement. Therefore we utilized the measured piezo waveform to time the acquisition of individual slices so that they were equally spaced along the scan axis (Fig. 2c, shown with 5 μm slice spacing). Notably the time intervals between slices are not uniform: slices near the extrema of the range are separated by longer intervals due to reduced scan velocity. Resonant galvanometer-based imaging systems exhibit similar nonuniformity in the angular velocity of the mirror, and these systems set pixel timing similarly[34]. Since piezo cycles are consistent after a brief initialization (Supplementary Fig. 1), a single measured cycle was sufficient to determine the timing of camera exposures throughout a multi-stack recording.

**Image-guided exposure timing**. However, we found that the sensor-guided approach to exposure timing was insufficient to yield images in the correct slice plane. Figure 2d (left panel) compares two images—one taken during fast scanning and the other taken statically—that were collected at nominally the same plane. Despite the fact that the measured piezo position was the same in both cases, there is a poor correspondence between the images. Initially we expected this to be explained by lag in the mixed analog and digital circuit that conveyed the piezo sensor signal. However to our surprise we could not explain the inaccuracy with a simple lag, gain, or offset of the sensor signal. We therefore developed a procedure to determine the correct timing for each image slice empirically by acquiring images at various temporal offsets from the naive sensor-based timing and choosing the optimal offset for each plane (Supplementary Fig. 2, methods). The right panel of Fig. 2d demonstrates that this image-guided refinement of the exposure timing corrects the alignment

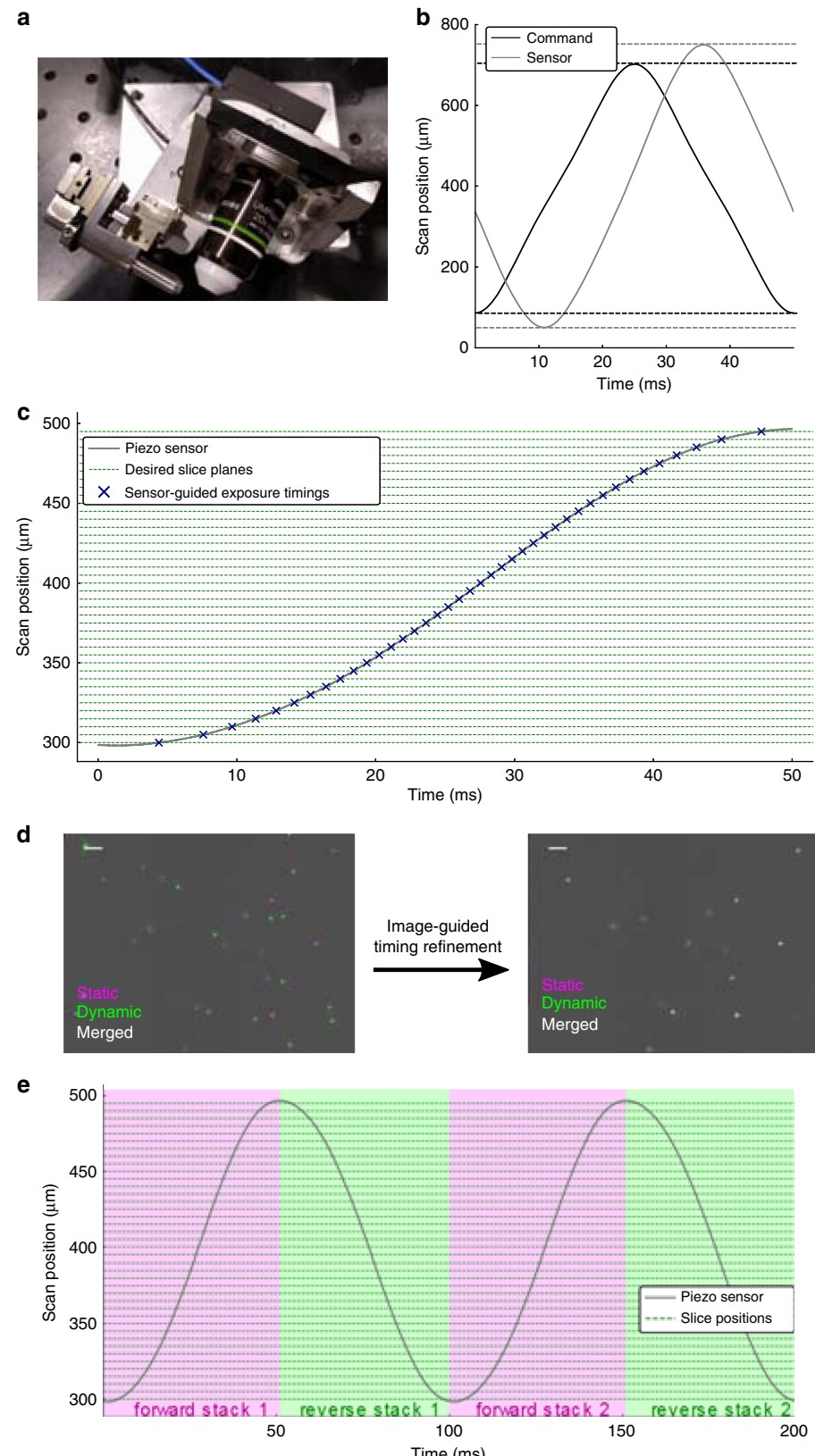

of the image planes. We found that the difference between the piezo sensor value and the true focal plane position was predicted by the acceleration of the scan system, suggesting that the error may be related to mechanical forces acting on the system (Supplementary Fig. 3).

**Pulsed illumination**. We also employed an optimized illumination protocol to achieve precise optical sectioning during fast scanning. Constant illumination is problematic for CMOS cameras operating with a rolling shutter because the start and stop time of the exposure differs for each row of pixels in a frame.

**Fig. 2** Improving the volume scanning rate of OCPI microscopy. **a** Photo of the scanning assembly (boxed region of Fig. 1b). Custom optics and machining minimize inertia. **b** Distortion of the piezo response is reduced relative to Fig. 1d by using a lowpass-filtered command. Dashed lines illustrate that the measured amplitude does not match the command. The command was optimized iteratively to generate the desired 700 µm range at a 20 Hz scanning rate. **c** A 10 Hz scan of a 200 µm volume overlaid with dashed lines marking depths at which to acquire images with 5 µm spacing between optical sections. When sensor output is used to guide image acquisition, an image is acquired at each intersection of the sensor trace with a dashed line (marked with x's). Note the uneven spacing of the x's along the time axis. **d** Sensor-guided image acquisition timing is insufficient to specify the correct image plane during a fast recording. A dynamically-acquired image of fluorescent beads is overlaid with an image taken when the scanner was stationary and, according to the piezo sensor, in the same image plane. An additional image-guided timing calibration corrected this inaccuracy and ensured that each slice of the stack was located in the correct plane (see Methods section, Supplementary Fig. 2). Scalebar: 5 µm. **e** Volume imaging rate was increased by an additional factor of two by imaging each plane twice per scan cycle. Thus a 20 Hz imaging rate is achieved for a scanning rate of 10 Hz. We compensated for non-simultaneous and non-uniform temporal spacing by interpolating each adjacent pair of stacks, resulting in virtual stacks aligned with the transitions between colored regions

Since the scan system is constantly in motion this implies that each row of pixels samples a slightly different axial plane. In order to prevent this contamination of an image with information from multiple axial planes we utilized pulsed illumination. By using a brief, well-timed pulse we were able to ensure that photon integration occurred only during the global exposure period during which all rows of pixels on the camera chip are exposed simultaneously (see Methods section).

**Bidirectional imaging**. Typical volume scanning systems acquire images while sweeping the scanner in one direction, but they do not acquire during the subsequent flyback segment of the command cycle. We chose instead to acquire images while sweeping in both directions. Since each depth in the volume is visited twice per command cycle, our volume scanning rate is 2× the command frequency. Thus we acquire a forward stack and a reverse stack within a single command cycle. Image-guided timing was optimized separately for slices of the forward and reverse stacks (Supplementary Fig. 2). Figure 2e diagrams four image stacks acquired in this manner during two scan cycles. Under this bidirectional paradigm the time interval between consecutive images is constant only for the plane in the center of the scan and becomes less uniform in planes closer to the extrema of the scan cycle. However the average sampling interval of all planes is a constant, equal to half the duration of the piezo command cycle. An equivalent statement is that bidirectional imaging doubles the average sampling rate.

Non-uniform sampling rate may affect the accuracy of timeseries analyses. More generally, timeseries analyses often consider each image stack as a single timepoint even though slices within each stack are acquired in sequence, not simultaneously (light field microscopy is an exception). We addressed both of these pitfalls by interpolating each consecutive pair of stacks in time, yielding stacks with a constant virtual sampling time corresponding to the lines of color transition in Fig. 2e (see Methods section). While inferior to truly simultaneous sampling, we expect that this correction will improve the fidelity of timing-sensitive analyses. Taken together with the scan capability shown in Fig. 2b, our system supports an average volume imaging rate of 40 Hz when scanning a 700 µm axial range (a scanning speed of 28 mm s$^{-1}$). This is 14× faster than previously demonstrated with OCPI or SPIM[32]. Thus, OCPI microscopy is no longer limited by scan rate when imaging medium-to-large volumes.

**Distributed planar imaging**. After overcoming the scan rate bottleneck, further speed improvements for LSFM must come from mitigating the camera frame rate bottleneck. Maximum readout rates of scientific CMOS cameras are all similar (Fig. 1f), and these rates have not improved within 6 years. Rather than waiting for faster cameras we devised DPI to parallelize

acquisition across multiple cameras by exploiting a feature of CMOS camera design: maximum frame rate depends on the size of only one dimension of the image (Image Height in Fig. 1e) and thus volume imaging rate scales with only two of the three image dimensions. Our strategy was to divide the image volume into two halves along the rate-limiting axis of the camera, and relay the halves to different cameras. Each half-image can then be imaged at twice the maximal rate that a single camera can capture the full volume (Fig. 3a). We cut the images by positioning the apex of a knife-edged mirror (KEM) in the focal plane of the tube lens, introducing a 90° fold along the center of the focal plane. The two halves of the focal plane were then relayed to the two cameras. The exposures of the two cameras were synchronized, and their images were later stitched back into a single image with custom software (see Methods section). If the apex of the KEM is not precisely in the focal plane of the tube lens then a strip of the image will be captured redundantly (but with reduced intensity) on both cameras. Figure 3b, c demonstrates in pseudocolor the alignment and overlap of stitched images of a fluorescent bead sample, revealing that beads along the edge of the KEM were imaged on both cameras. We aligned our system so that the redundantly imaged region was only 10 pixels wide, meaning that 99.5% of pixels sample an independent region of space when each camera exposes half of its available pixel region (Fig. 3d). We utilized an off-the-shelf KEM that exhibited roughness at the very edge of the mirror surface due to manufacturing limitations. This roughness scatters incident light and leaves a subtle stripe in the stitched image of a densely fluorescent sample, barely visible in the grayscale image of a larval zebrafish brain slice (Fig. 3e).

Multiple DPI modules can be chained to further divide the image and relay the partial images to additional cameras. Chaining yields a linear increase in frame rate with each additional camera. This increase comes at the cost of a modest linear increase in image redundancy and a decrease in photon efficiency with each additional camera, with the efficiency most sensitive to the transmission efficiency of the relay lens system (Supplementary Fig. 4). We estimate that a 16-camera system (nearly 16× speedup) with 90%-efficient relay lenses would have a total transmission efficiency of 60%(Supplementary Fig. 4c).

**Identification and removal of heartbeat artifact**. In order to demonstrate the impact of overcoming both the scanning and frame rate bottlenecks, we used the new system to record neural network dynamics in a zebrafish brain expressing GCaMP6f[1] (Fig. 4). We chose to image at ×10 magnification in order to maximize camera frame rate. Note that magnification (and field of view) can easily be changed by swapping in a different objective lens. We imaged a volume encompassing 40 slice planes in the forebrain ($223 \times 127 \times 200$ µm) of a zebrafish larva with a 20 Hz volume rate and $0.65 \times 0.65 \times 5$ µm voxel size over a

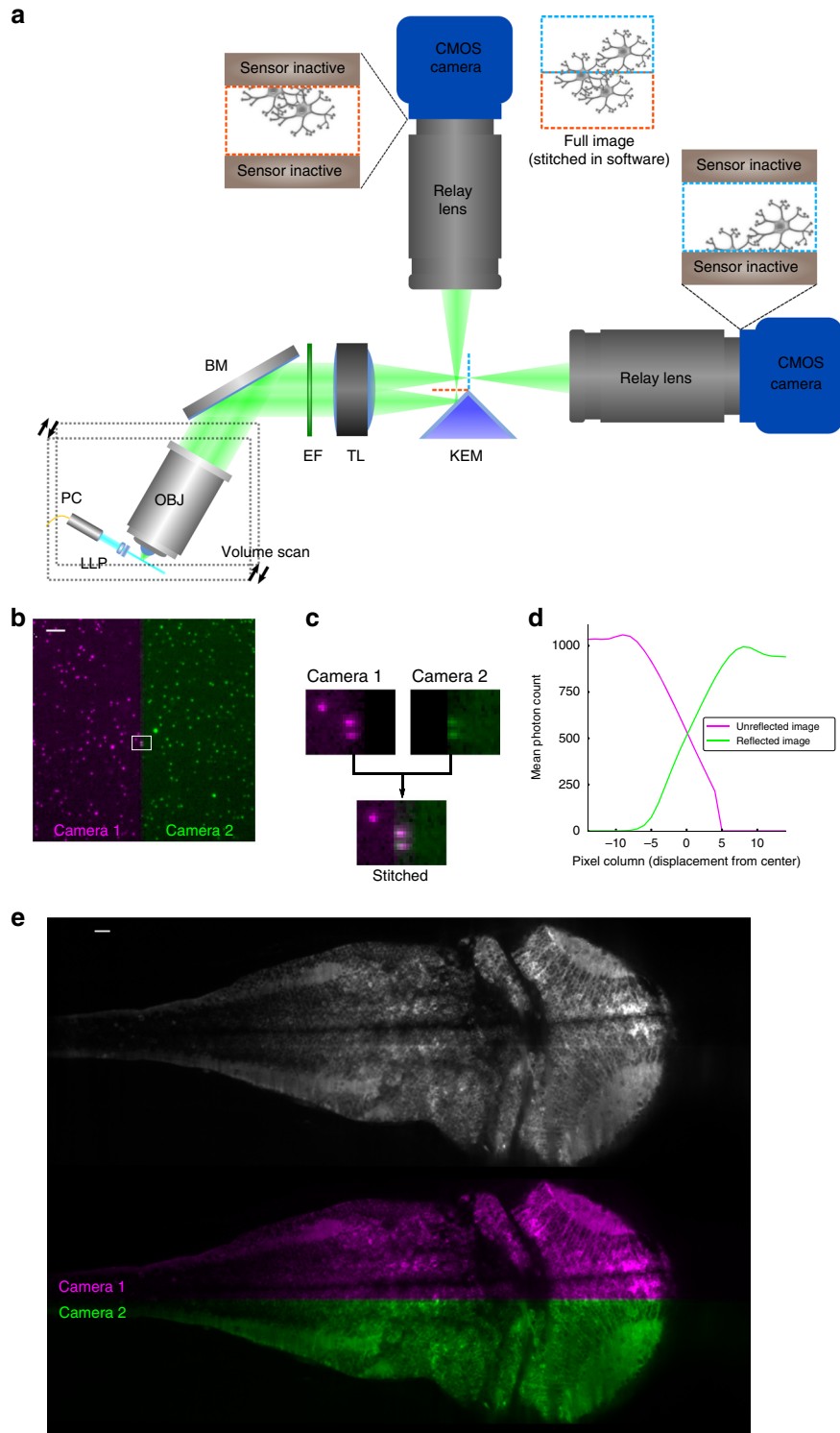

**Fig. 3** Mitigating the camera bottleneck. **a** A two-camera shared-image OCPI system. A knife-edged prism mirror (KEM) takes the place of the camera sensor in Fig. 1b. The mirror is aligned so that half of the image is reflected and relayed to a camera above while half passes unimpeded and is relayed to a second camera. Cameras are aligned so that they image a centered horizontal band in the field of view. The two cameras expose synchronously, and their images are later stitched together into a full image. Since the frame rate of a CMOS camera depends only on image height (Fig. 1e) this doubles the imaging speed of the system. **b** Example stitched image of fluorescent beads (0.2 μm diameter) with one camera's image in magenta and the other in green. Scale bar: 20 μm. **c** Zoomed view of the rectangular region marked in panel **b** showing a pair of beads in the narrow region imaged by both cameras, corresponding to the apex of the KEM. **d** Quantification of a stitched image of fluorescene dye solution with the same width and location as shown in panel **c**. The width of the redundant image region is approximately 10 pixels (less than 0.3% of the camera chip width). **e** Pseudocolored and grayscale views of the same stitched slice of a larval (5 dpf, *HuC:GCaMP6s*) zebrafish brain. Scale bar: 20 μm

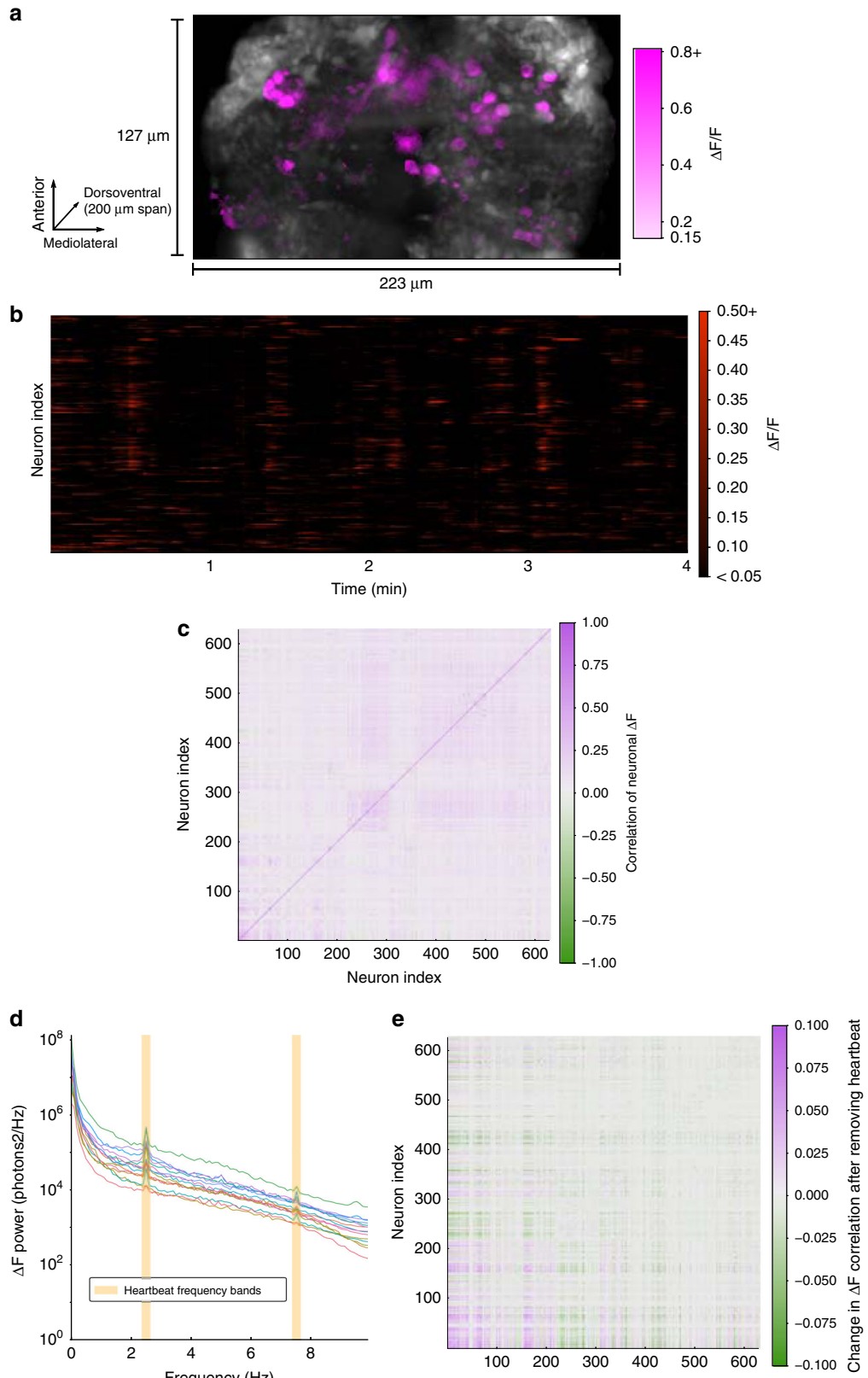

20-min period. In addition we imaged the whole brain of the fish (1020 × 348 × 200 μm) with the same voxel size at a rate of 10 Hz (Supplementary Movie 1, playback slowed 10×). Figure 4 focuses on the 20 Hz recording of the highly-active forebrain. Figure 4a shows a maximum intensity projection of the voxelwise change in fluorescence relative to baseline (Δ$F/F$) signal at a single

timepoint (see Methods section). Activity within individual slices is visualized in realtime in Supplementary Movie 2. We segmented a subset of 629 neurons by manually selecting regions of interest (ROIs), and we extracted Δ$F/F$ timeseries (see Methods section). A raster plot of cellular activity during the first 4 min of the recording is shown in Fig. 4b. Figure 4c shows a

**Fig. 4** 20 Hz imaging of zebrafish forebrain. **a** Maximum intensity projection of voxelwise $F$ (grayscale) and $\Delta F/F$ (magenta) along the dorsal-ventral axis of the larval zebrafish forebrain (5 dpf) with pan-neuronal *GCaMP6f* expression (*HuC:GCaMP6f*) acquired at 20 stacks/s at 10× magnification (Supplementary Movie 2). Only voxels with greater than 15% $\Delta F/F$ are colored. **b** Raster plot of $\Delta F/F$ within 629 manually segmented neuron ROIs over a 4 min period. Shown is an excerpt from a 20 min recording. ROIs were drawn smaller than the size of each cell in an attempt to minimize the effects of motion artifacts and cross-talk between nearby neurons. **c** Neurons exhibit a range of pairwise correlations in the $\Delta F$ signal. Correlations were computed with highpass-filtered neuron traces (1.0 Hz cutoff) in order to focus on relationships revealed by high sampling rate. Neurons are ordered by axial depth in the forebrain (dorsal to ventral). **d** Power spectra of the $\Delta F$ signals for a subset of neurons show that power diminishes gradually with increasing frequency. These and many other neurons exhibit peaks in their spectra at 2.5 Hz and 7.5 Hz that correspond with the larval zebrafish heart rhythm[35]. **e** Bandstop filters were applied to remove the heartbeat frequency bands (see Methods section) before recomputing correlations. Shown is the matrix of differences in correlation values obtained before and after heartbeat artifact removal ($corr_{after} - corr_{before}$). Thus this matrix highlights spurious correlations due to heartbeat that could contaminate a naive analysis of neuronal activity

commonplace analysis of neuronal timeseries, a matrix of pairwise correlations, computed over the entire 20-min recording. Before computing correlations, neuron traces were highpass filtered with a cutoff of 1.0 Hz in order to emphasize correspondences over short timescales.

Genetically encoded calcium indicators have long decay times, on the order of 400 ms for *GCaMP6f*, calling into question whether additional information is gained by increasing the sampling rate. However since the indicator rise times are much shorter (about 50 ms for *GCaMP6f*[1]) we hypothesized that higher sampling rates will be informative. In order to estimate the information gain from sampling at 20 Hz we computed the power spectral density (PSD) of the $\Delta F$ signal in each neuron. PSDs for 15 neurons are plotted in Fig. 4d. Indeed we found that power diminishes only gradually up to the maximum frequency (10 Hz) permitted by the Nyquist sampling theorem. Moreover we noted peaks at 2.5 Hz and 7.5 Hz in the spectra of many individual neurons. These peaks underlie correlations much larger in magnitude than neighboring frequency bands of the signal (Supplementary Fig. 5a), and the size of the 2.5 Hz peak is highly correlated with the size of the 5.0 Hz and 7.5 Hz peaks relative to other frequency pairs (Supplementary Fig. 5b). These attributes suggest that the peaks are frequency components of a signal with a fundamental frequency of 2.5 Hz, and that this signal is mixed into the signals of many neurons. The frequencies of the peaks lie within the range of frequencies found in a fluorescence-based motion-tracking study of the fish's beating heart[35]. Therefore we suspected that the peaks reflect motion artifacts induced by the beating heart of the fish. In order to explore this further we imaged a fish before and after administering a high dose of anesthetic (tricaine) to stop the fish's heart. In this fish we imaged a single plane at a higher sampling rate and observed peaks at a fundamental frequency of 1.8 Hz and the 3.6 Hz harmonic (Supplementary Fig. 6a). After administering anesthetic these peaks were fully attenuated (Supplementary Fig. 6b), strongly suggesting that this is a physiological artifact related to the heartbeat. Moreover the artifact is induced by specimen motion; pixel displacements found through image registration also exhibit peaks at 1.8 Hz and at higher harmonics (Supplementary Fig. 6c).

Such heartbeat artifacts could bias the results of analyses of neuronal dynamics, just as motion artifacts have been found to bias the results of human brain imaging studies[36]. In order to determine the effect of this artifact on the pairwise correlation measure, we first removed the artifact by applying bandstop filters to remove the heartbeat frequency bands from each neuronal signal. We then computed new pairwise correlations, and for each pair of neurons we recorded the change in correlation magnitude pre vs. post artifact removal (Fig. 4e). This analysis suggests that correlation values were overestimated or underestimated by as much as 0.1, highlighting the danger that the beating heart could induce spurious correlations into neuronal network analyses. We conclude that many future zebrafish neuroscience studies would

benefit from adopting sampling rates of at least 15 Hz (the Nyquist frequency of the highest observed frequency component, 7.5 Hz) so that the artifact can be filtered out (as we demonstrated) or else removed by regression of pixel intensity against the motion vectors found by image registration. The latter approach would be more robust to variation in the heart rate of the fish during recording.

## Discussion

OCPI is similar in principle to SPIM[4], one of the earliest LSFM implementations, but there are crucial differences. OCPI was the first implementation to achieve volumetric imaging by translating the optics instead of the sample[6]. OCPI also introduced a 30–45° tilt in the optics to facilitate observation of extended horizontal samples, such as neuronal tissue slices or in vivo preparations, while minimizing the path length of both the illumination and emission light through the sample. Finally, from the outset OCPI reduced weight and the geometric hindrances that would arise from having two objective lenses by generating the light sheet using custom optics. Note that the speeds of these techniques are more similar than they seem; most are camera-limited and the variation in demonstrated rates arises from differing choices in camera ROI size. However, OCPI was historically unable to achieve the volumetric scanning speeds demonstrated by several more recent LSFM techniques (1.1 mm s$^{-1}$, 20 mm s$^{-1}$, 4.4 mm s$^{-1}$, 2 mm s$^{-1}$, 2 mm s$^{-1}$, 3.2 mm s$^{-1}$, 5.3 mm s$^{-1}$)[10,15,17,23,26,27,32].

Our results demonstrate that improvements in hardware and software design allow OCPI microscopy to meet or exceed these scanning rates (28 mm s$^{-1}$ demonstrated) and also to achieve commensurate imaging rates provided that a sufficient number of cameras are used in a DPI configuration. Considering all tradeoffs (discussed below), we anticipate that fast OCPI will be a technique-of-choice for many experiments requiring maximal imaging speed of large volumes.

When compared to other fast direct-imaging LSFM variants, OCPI avoids aberrations caused by imaging outside of the native focal plane of the objective[10,27]. While some oblique imaging methods[8,22,23,28] avoid these aberrations, they are not photon-efficient: only a fraction of light collected by the objective reaches the image sensor. Efficiency is of central importance for neurophysiology as the field transitions from calcium to voltage indicators[2,3], which demand a much higher sampling rate and thus more imaging repetitions without inducing phototoxicity[37]. The highest efficiency demonstrated in an aberration-free oblique configuration is 21%[28]. Recently an efficiency of 73% was achieved in an oblique configuration[38] at the expense of aberrations: diffraction-limited imaging was confined to a $70 \times 20 \times 100\ \mu m$ region. OCPI utilizes all light collected by the objective lens and makes no compromise with regard to aberrations. Oblique systems also require a relatively high-NA objective, but moderate NAs are often desirable with LSFM: increasing NA improves light efficiency but also worsens aberrations due to

refractive index mismatches between tissue and media. Higher NA also implies reduced depth-of-field, which demands a thinner light sheet with a shorter Rayleigh range, corresponding with a smaller field-of-view[4]. Thus the optimal choice of detection NA is application-specific. We find that NAs of 0.3–0.5 trade favorably between efficiency, resolution, and field-of-view when imaging large numbers of neurons in living specimens. In cases where a different tradeoff is desirable our design allows for easy swapping of standard objective lenses. Future work could mitigate the field-of-view tradeoff by incorporating a propagation-invariant light sheet[39,40].

The original SPIM implementation[4], perhaps the closest relative to OCPI microscopy, shares the optical advantages of OCPI but scans by translating the sample rather than the optics. This is a crucial difference at high scanning rates: OCPI avoids jostling the sample, and it's possible to optimize the response of the scan system without regard for the mass of the sample being imaged. However OCPI scanning may still disturb delicate specimens, and thus may be incompatible with some mounting procedures[41]. Our rigid mounting procedure (see Methods section) limits access to the fish's body, but has the advantage that we need not paralyze the fish; we avoid swim motion artifacts because the fish rarely attempts to swim.

OCPI and SPIM also have the unique advantage that the size of the imaged volume is limited only by the range of the linear actuator, whereas other techniques are limited to a volume set by the field of view of the objective. This advantage is currently underutilized because the opacity of samples limits imaging depth. A two-photon light sheet[9] improves imaging depth, but in practice 2P LSFM does not offer the same depth advantage as point-scanning 2P. This is because with 2P LSFM, as with 1P LSFM, scattered emission cannot be attributed to a precise location in the sample[42]. Greater depth may also be achieved by incorporating structured illumination[43,44] or multi-view imaging[45] into future OCPI microscopes, but these solutions decrease imaging rate. Alternatively one could modify tissues to match the refractive index of the media (i.e., reduce scattering), but as of now this is only possible in fixed tissue[46–48]. Another promising direction for future work would be to alter the OCPI scan direction to be parallel to the sample surface, avoiding scanning deep in the tissue.

Outside of LSFM, computational techniques such as light field microscopy offer extremely fast 3D image acquisition, but the computational complexity of image deconvolution can be prohibitive, especially for lengthy imaging sessions. Relative to light field microscopy, direct imaging methods such as OCPI also exhibit a more favorable tradeoff between resolution and imaging rate, and they permit real-time analysis of imaging data[49,50].

Our scanning optimizations can accelerate OCPI, as well as any microscope that relies on mechanical scanning. In particular the optimizations will apply to LSFM variants that scan the light sheet using a galvanometer while synchronously scanning the detection objective[51]. The only caveat with this application is that synchronizing the illumination and detection scans may be challenging at high speeds. Galvo-scanning the light sheet also requires larger illumination optics, meaning that larger specimens (such as a behaving mouse) cannot fit under the microscope. OCPI's miniaturized static illumination path is better suited for these applications.

Our scan system was tuned manually in an attempt to simultaneously satisfy multiple scan rates, ranges, and amplitudes, and we conservatively utilized only a fraction of its 100 N maximum force output (Supplementary Fig. 3). In the future an automated procedure will allow one to choose optimal parameters for each recording session and to more easily swap in an objective lens with a different mass. The procedure should optimize the control

system so that the piezo response is closer to an ideal triangle wave. A triangle wave is optimal because image slices can be distributed uniformly in both space and time. When images are not spaced uniformly in time, as diagrammed in Fig. 2c, the camera spends a fraction of each stack idle and thus its maximum frame rate is not fully utilized. The achievable frame rate of the camera depends on the shortest interval between frames, 0.8 ms in Fig. 2c. When compared to the 1.25 ms interval that would be possible with a true triangle wave it is apparent that the camera spent 37% of the time idle. Future optimizations will prioritize reducing this idle time.

DPI addresses the bottleneck resulting from limits in camera frame rate, providing an increase in frame rate proportional to the number of cameras used. DPI can be integrated into any LSFM design (and more generally any widefield plane-to-plane imaging technique) because only the components downstream of the tube lens need to be modified. Techniques that do not image a plane to a plane may also benefit from DPI. For instance, light field microscopy could benefit by combining multiple camera sensors into one large virtual sensor to improve resolution and/or imaging rate. Parallel LSFM[52] (pSPIM) also combines multiple cameras to speed up imaging, but this technique is less scaleable and less general: pSPIM parallelizes image acquisition across a limited number of axially-distributed planes and requires deconvolution to recover focused images. Moreover pSPIM requires thin samples imaged with a tilt large enough to provide disjoint beam/sample intersections. An alternative approach[53] is to use burst imaging cameras that operate at much higher frame rates than the scientific CMOS cameras commonly used for LSFM. Unfortunately these cameras exhibit much greater noise and less efficiency (23 e$^-$ vs. 1.4 e$^-$ rms noise and 50% vs. 82% efficiency when comparing PCO.Edge vs. PCO.dimax cameras). Thus the faster camera requires ≈26× more light to achieve the same signal-to-noise ratio as the scientific camera. This inefficiency becomes prohibitive for longer recordings in demanding applications such as voltage imaging. Furthermore these cameras are limited to brief imaging sessions because the onboard camera memory fills faster than the images can be transferred to a computer.

Another artifact that can bias analyses is the striping artifact common to most LSFM implementations. Resonant scanning of the light sheet prevents the artifact[54], but newer approaches using static optics[55,56] are more compatible with the miniature illumination path of OCPI and will be integrated in future work.

Fast mechanical scanning and DPI establish a solid foundation for studying fast dynamic processes—such as signal transmission between neurons—at scale. We also demonstrated that a high sampling rate allows one to remove physiological artifacts such as heartbeat that could bias fluorescence timeseries analyses. We expect that heartbeat artifact removal will become a standard preprocessing step when analyzing zebrafish imaging timeseries. In combination with advances in fluorescent indicators and in large-scale image analysis, these improvements to microscope hardware bring us closer to a more comprehensive understanding of brain-wide activity.

## Methods

**Calculation of image quality in non-native focal planes**. See Supplementary Note 1.

**Theoretical resolution calculations**. Resolution expectations shown in Fig. 1a, c were calculated based on established Gaussian approximations of the PSF[57]. The theoretical axial light sheet PSF in Fig. 1c was found by multiplying two such Gaussian approximations: the axial detection PSF (set by the detection NA) and the illumination PSF at the waist of the light sheet (set by the illumination NA).

**PSF fitting**. The empirical PSF shown in Fig. 1c was found by first embedding multi-color fluorescent beads (0.2 μm diameter) in agarose gel and acquiring an image stack at 0.1 μm axial spacing (imaged in the 500 nm to 550 nm emission band). A custom algorithm then found bead locations based on local maxima in image intensity. Image ROIs around each maximum were extracted and a Gaussian function was fit to the axial intensity distribution. The axial offset of the Gaussian was a free parameter, allowing for sub-voxel estimation of the bead location. Fits more than 5× larger than the diffraction limit were assumed to be clusters/clumps of beads and excluded from analysis. Using only the remaining fits, the median PSF size was calculated within a 10 μm-wide sliding window (slid along the direction of light sheet propagation). The location of the waist of the light sheet was inferred as the offset where the windowed median was minimized. The 20 bead ROIs nearest to the waist were aligned using the fitted center coordinates, resampled at 0.1 μm isotropic spacing, and averaged to generate the image in Fig. 1c. The curve to the right of the figure is the measured axial intensity distribution of this average bead.

**Off-the-shelf components**. A laser system (Spectral LMM 5) output a collimated gaussian beam via a pigtailed fiber-optic collimator (1 mm to 3.3 mm diameter, variable via an attached iris). This beam was passed first through an achromatic doublet lens (either Edmund Optics 45–262 or 45–207) and then a cylindrical lens (see next section), and finally a coverslip before reaching the sample. The lenses were sealed inside a housing so that water could not enter into the lens space when submersed in the sample dish. The alignment of the light sheet was adjusted with a set of small stages (Elliot Scientific MDE266 and MDE269). Either the Olympus UMPLFLN10X/W, UMPLFLN20X/W, or LUMPLFLN40X/W objective collected emission from the sample. The light sheet and objective were mounted 60° from the horizontal axis and scanned together (Piezosystem Jena NanoSX800 piezo-electric positioner, 30DV300 amplifier). A stationary broadband mirror reflected the output from the objective horizontally to a 200 mm tube lens (Thorlabs ITL200) placed at the 1f distance from the objective's back focal plane. The KEM (Thorlabs MRAK25-G01) was placed in the image plane behind the tube lens, and sometimes swapped with a 50/50 beamsplitter (Thorlabs BSW10R) for alignment purposes (see below). Two telecentric relay lenses (Edmund Optics 62–902) relayed the divided image to the cameras (PCO Edge 4.2). Analog and digital I/O to the positioner, cameras, and laser was managed by a PCI data acquisition device (National Instruments PCI-6259) with a single sample clock shared across all signals. A PC with two RAID arrays (10 hard drives each) streamed the output of the cameras to disk (up to 1 GB s$^{-1}$ per camera at maximum frame rate). The sample was positioned on a physiology breadboard (Thorlabs PHYS24BB) mounted to a lab jack (Newport 281) and XY stage (Scientifica).

**Custom components**. The only custom optics were the small cylindrical lenses used to form the light sheet. Two different lens configurations were used to focus the light sheet either to a 5.3 μm waist (roughly matching depth-of-field of the NA 0.3 objective) or to a 2 μm waist (roughly matching depth-of-field of the NA 0.5 objective). The thicker sheet was formed by pairing Edmund Optics 45–262 with a custom cylinder lens (Tower Optical) with focal length of −6.25 mm and diameter of 3 mm. The thinner sheet was formed by pairing Edmund Optics 45–207 with Edmund Optics 48–373 (diameter of 48–373 was customized to 5 mm). Full lens specifications are included in Supplementary Note 2. Custom mechanical components were designed collaboratively and refined iteratively in collaboration with the Washington University Medical School Machine Shop. Hardware for the scan system was designed to minimize weight. A parts list, schematics, and labeled photos are also included in Supplementary Note 2. We found that the small dovetail stages used to align the lightsheet exhibited a few microns of motion in their joints when scanning at high rates. This motion defocused the image and required correction by modifying the dovetail slides to add a locking screw. A magnetic swappable Thorlabs filter cube insert (Thorlabs DFMT1) was modified to hold the knife-edged prism mirror. The 1″ apertures of the filter cube itself (Thorlabs DFM) were widened to 1.1″ with a standard boring tool to prevent vignetting of the relayed image.

**Calibration of piezo closed-loop controller**. Initial calibration of the piezo control system (NanoSX800 with 30DV300 amplifier) was performed by the manufacturer (Piezosystem Jena). We requested that they optimize the control system for the highest achievable frequency and amplitude of operation with a triangle wave command, 400 g load, and a translation angle of 30° from vertical. They tuned PID parameters as follows: $k_p = -0.3$, $k_i = 50$, $k_d = 0.1$. The load used in the experiments detailed in this article was smaller (264 g), so we further refined the calibration using the iterative procedure described in the product manual. One parameter at a time was manually adjusted by serial command, and the response of the system was measured. If a parameter update drove the system into oscillations, then the system was immediately switched to open-loop mode and the parameters reset. A detailed PID tuning procedure is available in the product manual. The final PID parameters after this secondary tuning were: $k_p = -0.37$, $k_i = 50$, $k_d = 0.11$.

**Generation of smoothed triangle wave commands**. A triangle wave with the desired frequency, amplitude, and offset was lowpass filtered with a cutoff of 3.25×

the triangle wave frequency (32.5 Hz for a 10 Hz triangle wave). This resulted in erosion of the triangle peaks and a reduction in the range of the command. In order to compensate for this reduction the original triangle wave was expanded and filtered again iteratively until the filtered wave matched the desired range.

**Scan range tuning**. The piezo command waveform was adjusted iteratively until the maximum and minimum values of the piezo cycle (as measured by its built-in sensor) matched those requested by the user. The initial guess for the command range was set to 10% smaller than the target range to guard against potential damage from overshoot. The piezo was then operated with this repeated waveform for a 20 s initialization period before measuring the sensor response for a cycle. The lower and upper limits were then updated independently based on sensor feedback with the same procedure: error was calculated as (target–measured), a value equal to 90% of this error was added to the limit used to generate the command signal, and a new command signal was generated. This was continued until both the upper and lower limits matched the user's request within a 0.1 μm margin of error. These stopping criteria were met within 5 iterations or less.

**Pulsed illumination**. Since the cameras operate with a rolling shutter, only the latter part of the exposure interval corresponded to simultaneous (global) exposure of all CMOS sensor lines. Laser pulses were timed to occur only within this global interval in order to prevent image information from bleeding into adjacent slices of the stack. When the camera is operated at maximum frame rate the duration of the global shutter period is only one line time (9.76 μs for PCO. Edge 4.2). At sub-maximal frame rates the global period is equal to the difference between the chosen exposure time and the shortest possible exposure time that the camera can sustain. Therefore one can prevent bleeding of image information into adjacent slices of a stack during dynamic recordings by using brief illumination pulses aligned with the end of each frame and operating the camera slightly below its maximum frame rate. We pulsed the illumination laser only during the last 5% of the exposure interval, which required that the camera operate 5% slower than its maximal frame rate. During the imaging session shown in Fig. 4 the exposure time was 580 μs and the excitation pulse duration was 30 μs. Peak laser power was set to 18 mW for an average sustained laser power of 0.864 mW. Detailed diagrams of the camera's timing system can be found in the PCO.Edge 4.2 camera manual online.

**Image timing calibration**. After sensor-based timing of images proved inadequate we adopted an approach utilizing the camera to perform further calibration of the timing of each image of the stack separately so that the images corresponded to the intended planes of the sample. First a ground-truth image stack of 0.2 μm fluorescent beads was acquired at a very slow (0.1 Hz) scan rate so that factors such as dynamic forces and lag in the sensor circuit did not affect the appearance of images. Then, a fast dynamic recording was performed in which each image in both the forward and reverse stacks was acquired with various timing offsets relative to the sensor-based timing. The search space of timing offsets ranged from 0 μs to 1.2 μs at 50 μs intervals. Each dynamic slice image was then compared with the corresponding ground-truth static image and scored by similarity. The timing offset that produced the highest similarity score for a slice was chosen as the corrected timing for that slice. We noticed that fast dynamic operation produces not only an axial shift in each slice but also a lateral shift of less than 2 μm that varied by slice (likely due to compression and flexion of components). Therefore in order to calculate the similarity score we first performed 2D rigid image registration to shift the trial image laterally into alignment with the ground-truth image. After alignment the similarity score was calculated as the sum of squared differences between each pixel in the trial image and the ground-truth image normalized by the sum of squared intensity of pixels in the overlap region between images. Both the optimal temporal offset and the optimal lateral shift were recorded for each slice in the forward and reverse stack and used to acquire and align images of the zebrafish specimen. When using DPI, only the camera receiving the unreflected image was used for this alignment procedure.

**Interpolation of bidirectional images**. Bidirectional image acquisition produces image slices that are not uniformly spaced in time. A uniform sampling rate was simulated by interpolating new slices from each forward and back slice pair in the timeseries. Each interpolated timepoint was midway between the sample times, so each interpolated pixel intensity was simply the mean of the corresponding intensities of the two slices. This method also emulates simultaneous sampling of each image slice in the stack.

**KEM alignment with focal plane for DPI**. The camera receiving the unreflected image was aligned first using a procedure common to any OCPI microscope. The knife-edged mirror was installed in a modified magnetic filter cube insert so that it could be swapped easily with a 50/50 dichroic mirror (see image stitching method). The filter cube was incorporated into a cage system to maintain its alignment with the tube lens. This knife-edged mirror insert was placed in the cube and the distance between the tube lens and the cube was set by translating the cube along the cage axis and observing a dense bead sample on the camera receiving the unreflected image. The cube was approximately aligned when the image of the bead

sample on this camera was in focus and centered on the aperture of the cube (this also required lateral repositioning of the camera). A finer alignment of the cube was achieved by observing closely on both cameras the strip of the image corresponding to the knife edge of the mirror: the farther the edge of the mirror is from the relay image plane the larger the region of the image that is redundantly imaged on both cameras. Alignment was complete when the width of the redundant image region was minimized (Fig. 3).

**Camera alignment for DPI**. Since a row of pixels on the camera sensor is only 6.5 μm wide, slight rotations of the relative image planes of the two cameras can result in misalignment of the knife-edge line with the rows of the camera. Therefore in addition to the focal alignment above we also adjusted the relative rotations of the two cameras until the imaged knife edge line was horizontal. Since the KEM was mounted in a highly-repeatable kinematic insert, it was only necessary to perform this angular adjustment once.

The following procedure was used whenever the vertical size of the camera's active pixel region was changed: With the KEM installed, the active pixel region of both cameras was set as desired (settings on both cameras must match). Both cameras were activated to stream live updates of an image of a dense fluorescent bead sample. The camera receiving the unreflected image was translated up so that the KEM edge in its live image feed corresponded with the bottom edge of the pixel region. The other camera was likewise translated laterally so that the KEM edge aligned with the bottom edge of the pixel region. Since the image is reflected this edge corresponds to the top side of the unreflected image. For this reason all reflected images were flipped in software before performing the alignment steps described in the next section.

**Stitching DPI images**. After aligning the cameras and setting the desired pixel region, both cameras recorded an image of the beads simultaneously. Then the KEM was swapped with a 50:50 plate beamsplitter (Thorlabs BSW10R) mounted in a kinematic filter cube insert, the pixel region of the camera receiving the reflected image was set to full size ($2060 \times 2040$ pixels), and another image was recorded. It was critical that the bead sample remained stationary during the interim between these image snapshots. The full-size image spans the region where the two smaller images meet, and thus contains the information needed to align and stitch the smaller images. The bead sample was then replaced with the sample of interest, and the full imaging session was completed.

After all data were recorded, an image transformation was found to align the smaller bead images, and this same transformation was applied offline to stitch all images recorded in the zebrafish specimen. This transformation was found with the following 3-step procedure. First, a rigid 2D transform was found to align the smaller reflected image with the full image. This transformation was minimal in magnitude because both images were acquired with the same camera and centered on the same region of space (only the reflective surface was different). Second, a 2D affine transformation was found to align the full image with the smaller reflected image. A full affine transformation was allowed because subtle differences in alignment or manufacturing of the cameras and relay lenses cannot be captured by a rigid transform. The non-rigid component of the transformation was small (less than 1% scaling factor) but important to maintain alignment of beads throughout the image. In the third and final step the two transforms were composed into one transform (rigid first, affine second). This composite transform was applied to each reflected image of the zebrafish specimen to align it with the unreflected image. Before combining the two images the camera's constant bias intensity was subtracted from each pixel (by design the black level is not zero but a constant value).

**Zebrafish imaging**. *HuC:GCaMP6f* and *HuC:GCaMP6s* zebrafish larvae[58] were crossed with Casper larvae[59] for two generations to obtain transparent fluorescent larvae for imaging. Embryos were raised at 28.5C, screened for green fluorescence at 3 dpf, and imaged at 5 dpf. All zebrafish imaging was performed with the ×10 NA 0.3 imaging objective and 5.3 μm-thick light sheet. The larva was transferred by pipette into a drop of 1.5% low-melting-point agarose gel while the gel was still warm and in a liquid state. A syringe was used to suck the larva tail-first into a segment of Fluorinated Ethylene Propylene (FEP) tubing, and the gel was allowed to solidify. FEP was chosen because its refractive index closely matches that of water, and therefore aberrations are minimized when imaging through the material. The tubing segment was then mounted in a custom water-filled chamber at an angle of 60° from horizontal so that the rostrocaudal axis aligned with the light sheet. Excess tubing in front of and above the fish's head was cut away with a razor blade. All protocols were approved by the Institutional Animal Care and Use Committee at the Washington University School of Medicine.

**Stripe removal**. As is common with LSFM, we observed stripes in the zebrafish images due to scattering, absorption, and interference as the light sheet propagated through the sample. We applied a destriping filter to attenuate these stripes as a preprocessing step for the images in Figs. 3e and 4a, and Supplementary Movies 1 and 2. Each image slice was first log-transformed to account for the multiplicative (rather than additive) nature of the stripes. Next the slice was Fourier transformed,

and the magnitude of bins corresponding with the angle of the stripes was attenuated until the stripes were no longer visible when the image was reconstructed with an inverse Fourier transform. We did not apply the filter when performing timeseries analyses due to concern that the filter could introduce subtle artifacts. Most of the stripes observed were static and therefore could not be expected to influence the $\Delta F/F$ calculation. Brain regions exhibiting dynamic stripe patterns (for example regions behind the motile cilia of the olfactory rosettes) were not analyzed.

**Image registration**. The zebrafish, embedded in agarose and FEP tubing, moved little during the recording. However image registration was required to compensate for slow drift of the specimen. A single image stack was chosen as a fixed reference stack and each of the other stacks in the timeseries was registered to this stack. A custom algorithm found the simple shift (translation) that maximized the overlap between the fixed and moving stack.

**Extraction of neuronal calcium traces**. Neuron ROIs were selected by hand using a graphical software tool. A single 2D rectangular region was marked for each neuron in the plane that best appeared to capture the neuron's activity. As expected with LSFM, motion of cilia and blood cells in the fish's body induces time-varying stripe artifacts along the light sheet propagation axis in some image regions. We avoided segmenting neurons in these regions. The outer edges of the cells were excluded from ROIs in order to minimize crosstalk in the calcium traces of nearby neurons and to reduce the effect of motion artifacts. For each timepoint of each ROI the raw fluorescence value was calculated as the sum of voxel intensities within the ROI. $\Delta F/F$ was calculated by subtracting baseline fluorescence from this raw value and dividing the result by baseline. Baseline for each timepoint was calculated as the moving average of raw ROI fluorescence during the 60 s interval centered on the timepoint (the first and last 30 s of the recording were not analyzed). A different procedure was used for the analysis in Supplementary Fig. 6 (see "Anesthesia experiments" below).

**Anesthesia experiments**. Two 1-min-long calcium imaging sessions were performed in a single plane of the larval zebrafish forebrain at 100 Hz frame rate. A high dose ($1.3 \, \mathrm{g \, L^{-1}}$) of tricaine was administered after the first recording session, and a second recording of the same plane was taken 15 min later. PSDs were computed from the time-varying intensity within each image ROI, and the plots in Supplementary Fig. 6 show averages across all ROIs. All ROIs were 5 μm square regions, roughly matching the size of a neuron soma. In order to avoid any bias due to hand-selection of ROIs, all possible ROIs within a 130 μm square region were included in the analysis: PSDs were computed in a sliding window.

**Power spectral density and filtering of neuronal calcium traces**. All PSDs were computed with the Welch method with a rectangular window function on baseline-subtracted fluorescence traces. When filtering the signals to remove heartbeat frequency bands a Butterworth filter of order 5 was applied for each stopband. The filters were applied in both the forward and reverse directions in order to preserve phase.

**Reporting summary**. Further information on research design is available in the Nature Research Reporting Summary linked to this article.

## Data availability
The datasets generated during and/or analyzed during this study are available from the corresponding author on reasonable request.

## Code availability
We wrote several software modules to accomplish microscope control, PID adjustment, image timing calibration, image stitching, PSF measurement, temporal interpolation, and manual cell segmentation. Web links to these modules on Github are collected here: https://github.com/HolyLab/FastScanningAndDPI.

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

## Acknowledgements

We thank Zhongsheng Guo, Dae Woo Kim, and Ben Acland for their work in developing imaging software and the Washington University Medical School Machine Shop for their help in designing and machining custom optomechanical parts (John Kreittler, John Witte, and Kevin Poenicke). We also thank the users and staff of the Washington University Center for Cellular Imaging for consistent feedback that enabled us to debug and improve the microscope. Finally, we thank Xiaoyan Fu, Donghoon Lee, and Terra Barnes for comments and suggestions regarding the paper. This work was supported by NIH grants 1R24NS086741-01, 5R01NS068409-08, and 1T32NS073547-01.

## Author contributions

C.J.G. and T.E.H. together conceived the work and wrote the paper. C.J.G. built the microscope and acquired the data.

## Competing interests

T.E.H. has a granted patent on OCPI microscopy. C.J.G. declares no competing interests.
