## [Peer Review File · Nature Communications]

Reviewers' comments:

Reviewer #1, an expert in light sheet development (Remarks to the Author):

The authors provide a compelling case for their enhanced OCPI microscope and its capacity for rapid functional imaging in zebrafish larval brains. Essentially the revised OCPI, allows high-speed light sheet imaging via fast mechanical objective scanning and parallelization of the wide field acquisition across a pair of cameras. Typically, calcium imaging is performed at slower frame rates owing to the relatively slow calcium dynamics during neuronal firing. Nevertheless, the authors suggest that faster imaging is helpful to remove periodic artifacts in the data owing to the beating heart of the fish. The manuscript is well presented while the optimizations of the method, particularly the consideration of spatial and temporal sampling, are very rigorous and detailed. However, there are a few areas requiring some work before publication should be considered:

1) The authors correctly note that the fast light sheet techniques reported to date sacrifice spatial resolution and/or optical efficiency to achieve the fast volume scan rates with some also requiring a computationally intensive and artifact prone deconvolution. The analysis of figure 1 a) considers the diffraction limited range for refocusing beyond the normal working distance of a given objective as a function of NA. First the NA 0.4 line is not hugely useful since this is not a commonly available NA for a commercial objective. NA 0.3 would be a better choice, in keeping with the author's Olympus 10x/0.3w. Likewise NA 0.6 is rarely seen, and NA 0.8 would again be more representative of the authors objective choices (Olympus 40x/0.8*) and actually illustrate their point better still.

*I wonder if this is actually not the objective used since figure 2 a shows the 20x/0.5?

However, for NA 0.3 (suitable for cellular level resolution and often used for similar fast focusing light sheet systems), the diffraction limited refocus range will actually be rather large and certainly larger than the typical depth penetration achievable into tissue (from the NA 0.4 analysis one would expect somewhere around 5-600 microns). As such, the strength of the mechanical objective refocusing is contingent on its use at higher NA. Since light sheet methods are so light efficient anyway, the primary reason for doing so would be to improve the spatial resolution. The light efficiency is illustrated by the fact that the authors could afford such a short duty cycle for laser exposure. The authors note a lateral resolution of 0.65 microns, presumably based on sampling (6.5 micron pixels at 20x magnification, with Nyquist sampling). However, the case would be more compelling if there was some demonstration that diffraction limited resolution was achievable at NA 0.8 for example. PSF measurements would help in this regard. Nevertheless, the authors note the benefits of low NA for robustness to aberrations and a more appropriate overlap with the thicker illumination PSF. This does raise the question as to why it is necessary to utilize higher NA in the first place for such applications, when the performance in tissue will actually be poor and weakens the argument against existing methods.

2) The camera parallelization is certainly interesting, but similar (not identical but using knife edge mirrors and multiple cameras to split image across multiple cameras) has previously been reported by Dean et al. (Optica, 2017) for fast parallelized acquisition in light sheet microscopy. At the very least, this paper deserves mention and raises questions regarding the technical novelty.

3) The authors note that OCPI obviates the need for sample scanning. However, moving an objective in the same fluid as the sample can be similarly problematic. While this is fine for fish rigidly encased in agarose and a surrounding FEP tube as reported. More gently mounted samples or those that are secured only via the head will certainly be perturbed by this method. It would be good to mention this as a drawback and cite Vladimirov, Nature Methods, 2014 in this regard. This is also where motion free techniques such as OPM, ETL-SPIM, SPED and wavefront coding draw great benefit.

4) The OCPI method was reported early in the history of modern light sheet microscopy but to my

knowledge has not been adopted by others. The objective coupled cylindrical lens remains an interesting choice but there is little reasoning for it as described. For example, a galvo mirror could be used to scan a light sheet in synchrony with the detection optics (and at much higher speeds if necessary). This could either be achieved with a static cylindrical lens or an objective lens small enough to be located at 90 degrees, while allowing the detection objective to be mechanically refocused. In turn, this would also allow multi-color illumination with light sheets focused to the same depth and/or light sheet pivoting (using additional scanning optics) as desired. Since the authors report striping as a major concern, the method seems to overlook what could be done to easily address these artifacts, which are likely to be a greater source of difficulty than the beating heart. In this regard, the authors should cite Huisken et al., *Optics Letters*, 2007. Using the galvo method may even alleviate the difficulties of fast scanning owing to reduced mass and components providing drag. It would be good if the authors could explain the benefits of their approach thoroughly.

5) The authors state "To our knowledge the fastest volume imaging demonstrated with OCPI or SPIM was 4 Hz, and this was with only a 32 μm scan range" Lemon et al. *Nature Communications* 2015, achieve 5 Hz imaging of the entire drosophila central nervous system and with the benefits of multi-view imaging for improved optical coverage. This was using DSLM but is equally applicable to SPIM (the speed is not determined by the illumination - in fact DSLM is substantially more challenging in this regard.)

6) The authors state: "While "oblique" imaging methods (those that create the excitation sheet with the imaging objective 8,22,26,28) avoid these aberrations". However, oblique methods do not avoid aberrations, there is substantial coma and a reduction of axial resolution owing to the non-orthogonal illumination-detection PSF overlap as shown in the supplementary information of Bouchard et al. *Nature Photonics*, 2015.

7) The authors state: "Maximum readout rates of the top scientific CMOS camera manufacturers are all similar (Figure 1e), and these rates have not improved within 6 years, suggesting it may not be fruitful to wait for faster cameras." This is the case for sCMOS cameras. However, Mickoleit et al., *Nature Methods*, 2014 used an ultrafast camera to image at nearly 5000 fps and 60 vol/s accordingly. This should be noted.

8) The authors state "When compared to the 1.25 ms interval that would be possible with a true triangle wave it is apparent that the camera spent 37% of the time idle" this suggests that with 2 cameras the speed is 126% of a single camera operating at its full rate when combined with a faster focusing method. This detracts from the argument of parallelizing cameras before really solving the problem of fast focusing. Nevertheless, the authors have gone to some lengths to report where the method could be improved and the painstaking steps they have gone to to ensure that the system works as well as it does.

9) In general, regarding the assertion that artifacts resulted from the fish beating heart with a primary frequency at 15 Hz. It is unfortunate that this is only illustrated through lower frequency harmonics. Could the authors also illustrate this directly by imaging a single plane very fast to capture the 15 Hz with necessary Nyquist sampling (i.e. >30 Hz)? Also, it is simple to stop the heartbeat with excess anesthetic. This will change the neural activity for sure but then the loss of the oscillation can be shown in the power spectra.

10) Another reason to image faster is to capture voltage dynamics rather than calcium as a proxy. The voltage spiking would be much faster and the signals dimmer, justifying the choice of LSFM as an imaging technology and the higher speeds accordingly. While I would not expect the authors to provide such data at this point, mentioning it as a future direction would strengthen the arguments.

Reviewer #2, an expert in volumetric imaging of zebrafish brain (Remarks to the Author):

Report on Greer et al.

In this article, the authors report on technical developments aiming at maximizing volumetric imaging rate using OCPI. They specifically address two bottlenecks that they identified as currently limiting data throughput in this technique. The article offers several novelties or refinements compared to existing methods:

- 1) In order to increase the volume scanning rate, they optimize the closed-loop control command of the piezo holding the imaging objective. They developed an empirical calibration methods to precisely assign each temporal offset with a particular z-position.
- 2) They used a pulsed illumination approach to image well defined sections even during continuous z-scanning.
- 3) To go beyond the intrinsic limits of sCMOS camera framerate, they split the image in two, and used two cameras to image both halves, which are later stitched together.

Although I think that these constitute interesting developments, which should eventually warrant publication, I feel that the article, in its current form, does not provide the reader with a clear enough view of the gain offered by these methods. This is due to the fact that the performances of existing systems and of this new method are sometimes not correctly assessed. I thus list below a number of issues that should be addressed before publication.

- The authors discuss one of the important alternatives to mechanical scanning, i.e. remote focusing. They argue that this approach suffers from two major drawbacks: a partial loss of photons, and a degradation of the axial resolution. For the latter, I do not understand the line of reasoning as detailed in the supplementary materials: I feel that the calculation is flawed. It assumes that the intermediate optical system does not have a variable focal length, such as to successively bring the different z-planes in focus, as is precisely the principle of remote focusing. Without such possibility, the imaged formed on the sensor is obviously blurred as soon as the point source is away from the focal plane.
- Current state-of-the-art volume scanning rate should be indicated more clearly. The limitation is not in the scanning rate per se, but in the scanning speed (there is always a trade off between axial range and scanning rate). The authors should indicate what is the maximum speed in scanning that is currently attainable, and how it compares with the scanning speed that they do reach with their approach ?
- The authors define the volume imaging rate as twice the scanning rate, because they image each section twice during a complete scan (bidirectional imaging). I think this is misleading and that the authors should use a conservative definition of the imaging rate as the inverse of the largest time interval between successive images of any given slice. As I understand it, their imaging approach leads to uneven time intervals between successive acquisitions. For the sections located at the very top or bottom of the volume, the actual imaging period thus corresponds to the scanning period (and not the scanning half-period). The maximum volumetric imaging rate is thus 20Hz over 700um (which is already great !) but not 40Hz as claimed.
- When applied on zebrafish, the authors should indicate the total volume, inter-layer distance, and resolution for the large volume recordings at 10Hz (as is done for the smaller region ("In addition we imaged the whole brain of the fish at a rate of 10 Hz »)

Minor points :

- In the discussion, the authors write : « However, OCPI was historically unable to achieve the volumetric imaging speeds enjoyed by several more recent contributions ». Could they provide references and details of these recent contributions and how their performances compare with the present technique.
- They also indicate that « SPIM, perhaps the closest relative to OCPI microscopy, shares these optical advantages and differs only in that the sample is translated rather than the optics ». I think that SPIM does not necessarily involve the motion of the sample. Vertical scanning is in fact generally performed by displacing the laser-sheet with a galvo. So I think this remark should be removed.

Response to feedback from Reviewer 1 (Our responses are in bold font)

1) The authors correctly note that the fast light sheet techniques reported to date sacrifice spatial resolution and/or optical efficiency to achieve the fast volume scan rates with some also requiring a computationally intensive and artifact prone deconvolution. The analysis of figure 1 a) considers the diffraction limited range for refocusing beyond the normal working distance of a given objective as a function of NA. First the NA 0.4 line is not hugely useful since this is not a commonly available NA for a commercial objective. NA 0.3 would be a better choice, in keeping with the author's Olympus 10x/0.3w. Likewise NA 0.6 is rarely seen, and NA 0.8 would again be more representative of the authors objective choices (Olympus 40x/0.8*) and actually illustrate their point better still.

*I wonder if this is actually not the objective used since figure 2 a shows the 20x/0.5?

We fully agree with the reviewer's assessment that figure 1 a) would be more effective if it showed a different set of NA values. Therefore we have changed the figure to show values matching the popular UMPLFLN/LUMPLFLN-W line of Olympus objective lenses: (NA=0.3, M=10), (NA=0.5, M=20), and (NA=0.8, M=40).

***All zebrafish imaging shown in this manuscript was acquired with the 10x, 0.3 NA objective lens. However we also use the 20x, 0.5 NA objective regularly; swapping objectives is trivial due to the system design and the matched parfocal distances in this line of objectives. Our new PSF measurements (discussed below) are made with the 20x, 0.5 NA objective.**

However, for NA 0.3 (suitable for cellular level resolution and often used for similar fast focusing light sheet systems), the diffraction limited refocus range will actually be rather large and certainly larger than the typical depth penetration achievable into tissue (from the NA 0.4 analysis one would expect somewhere around 5-600 microns). As such, the strength of the mechanical objective refocusing is contingent on its use at higher NA. Since light sheet methods are so light efficient anyway, the primary reason for doing so would be to improve the spatial resolution. The light efficiency is illustrated by the fact that the authors could afford such a short duty cycle for laser exposure. The authors note a lateral resolution of 0.65 microns, presumably based on sampling (6.5 micron pixels at 20x magnification, with Nyquist sampling). However, the case would be more compelling if there was some demonstration that diffraction limited resolution was achievable at NA 0.8 for example. PSF measurements would help in this regard. Nevertheless, the authors note the benefits of low NA for robustness to aberrations and a more appropriate overlap with the thicker illumination PSF. This does raise the question as to why it is necessary to utilize higher NA in the first place for such applications, when the performance in tissue will actually be poor and weakens the argument against existing methods.

We agree with the reviewer (and our revised Figure 1a now shows) that the NA=0.3, M=10 objective lens performs well over a large axial range in the presence of aberrations induced by remote focusing. Currently this range (770 μm) is not a limiting factor because other technical limitations prohibit application of LSFM to larger living samples. However if future techniques address these limitations then the expanded scan range will find many applications (whole-brain imaging in mouse, for example). Therefore we think that our description of the remote-focusing problem and how to circumvent it may guide future microscope development, even for a system NA of 0.3.

We also agree that light efficiency is not a limiting factor in the specific imaging conditions shown in the paper. However we believe that light efficiency remains a critical consideration and thus larger NA values are useful even for a light sheet microscope, for two reasons:

1) **With current calcium indicators photobleaching and phototoxicity become significant when the recording session is extended. While using the NA=0.3 objective to image zebrafish we have observed significant dimming and reduced neural activity when the session is extended to 1 hour rather than the 20-minute session shown in the manuscript. Some experiments may only be tractable with longer recording sessions (circadian, developmental, and functional connectomic studies all being relevant examples). We**

also note from experience that HuC-driven fluorophore expression in zebrafish is quite high relative to many other combinations of driver and animal model where photons are more precious.

2) As the neuroscience community moves forward from calcium indicators to voltage indicators of neural activity, the required imaging rate (and thus photon flux) will increase dramatically. Typically the desired voltage imaging rate is 1000Hz, which is 50x faster than the rate that we demonstrated in the manuscript. If the stability and efficiency of the voltage indicator is comparable to GCaMP6f (currently a generous assumption), then one would expect a maximum recording duration of only ~1 minute.

It seems that the reviewer agrees with the latter point given a later suggestion to discuss voltage imaging. Thus it is apparent that we did not sufficiently highlight the utility of our microscope for voltage imaging. We have revised the manuscript to clarify our forward-looking approach and to emphasize the importance of light efficiency.

The reviewer suggestion of PSF measurements at higher NA is well taken. We have modified figure 1 to include PSF measurements of our system with the NA=0.5, 20x objective lens. While in principle the system can perform well at even higher NA, we argue that NA=0.5 is an optimal choice for many applications of interest. We obtain useful gains in resolution, photon efficiency, and axial scan range relative to aberration-limited methods (as shown in Fig. 1 a). At the same time this moderate NA balances the advantages of low NA that the reviewer mentions: 1) A depth-of-field matched light sheet can have a longer Rayleigh range (translating to a larger field of view), and 2) robustness to aberrations. We also note that in the future the system could be modified to mitigate these tradeoffs by using a propagation-invariant light sheet (Bessel or Airy) instead of conventional Gaussian optics.

2) The camera parallelization is certainly interesting, but similar (not identical but using knife edge mirrors and multiple cameras to split image across multiple cameras) has previously been reported by Dean et al. (Optica, 2017) for fast parallelized acquisition in light sheet microscopy. At the very least, this paper deserves mention and raises questions regarding the technical novelty.

We thank the reviewer for reminding us of this article. While their implementation relies on specialized preparations and geometries (specifically, thin samples imaged with a tilt large enough to provide disjoint beam/sample intersections), we agree that this is an important precedent and we have cited it in the revised manuscript. To more accurately highlight the specific features of our approach, we have changed the name from Multi Camera Image Sharing to Distributed Planar Imaging (DPI). The new name better communicates that this is a plane-to-plane imaging technique, emphasizes the parallel nature of the approach, and reflects the breadth of application of the technique to any widefield method. In the revised manuscript we contrast this with the parallel LSFM technique introduced in Dean et al 2017: pLSFM is a volume-to-volume imaging technique, applies only to light sheet microscopes, and requires deconvolution to recover focused images. Despite the noted similarities between DPI and pLSFM (camera parallelization and usage of knife-edged mirrors), we believe that these differences establish DPI as a distinct and novel technique.

3) The authors note that OCPI obviates the need for sample scanning. However, moving an objective in the same fluid as the sample can be similarly problematic. While this is fine for fish rigidly encased in agarose and a surrounding FEP tube as reported. More gently mounted samples or those that are secured only via the head will certainly be perturbed by this method. It would be good to mention this as a drawback and cite Vladimirov, Nature Methods, 2014 in this regard. This is also where motion free techniques such as OPM, ETL-SPIM, SPED and wavefront coding draw great benefit.

In light of this comment we have added a brief discussion of the advantages and disadvantages of our specimen mounting technique, and also added a citation of Vladimirov et al 2014. The advantages of our technique are 1) As the author stated, the rigid tubing is robust to perturbation that may be induced by the motion of the objective lens in the media, and 2) We avoid paralyzing the specimen. We observe that

the rigidity of our mounting discourages swim behavior, thus reducing motion artifacts that would otherwise contaminate recordings of unparalyzed fish. (We recognize that this may be *undesirable* in a study of swim behavior, but would be beneficial in many other experimental paradigms because we eliminate off-target effects of the paralytic in the nervous system). The disadvantage of our approach is, as the reviewer stated, that it is likely incompatible with some existing mounting techniques, especially those used to study motor function.

In our view it is not clear how well our fast scanning approach will work with a gentler mounting method. The working distance of the objective, angle relative to the water surface, scan rate, and the size/shape of the imaging chamber may all play a role. Anecdotally we have reason to be optimistic; we observe that the effects of motion in our setup seem confined to a surprisingly superficial portion of the water column. We haven't systematically explored this, but one explanation is that the ~2cm depth of water in our chamber places us in the "deep water" regime, in which the effects of waves along the surface decay exponentially with depth in the water column (See equations for horizontal and vertical particle excursion here: https://en.wikipedia.org/wiki/Airy_wave_theory). As this is speculative and non-trivial to verify we have not mentioned it in the manuscript.

4) The OCPI method was reported early in the history of modern light sheet microscopy but to my knowledge has not been adopted by others.

While the name has not been adopted by others, we are not sure we agree with this assessment. OCPI was the first to perform rapid volumetric imaging by scanning the optics rather than moving the sample; OCPI was the first to image horizontal samples by introducing the illumination at a 45° angle (though iSPIM, which came several years later, is often credited with this); and OCPI was the first application to calcium imaging. These are all elements that have been widely disseminated.

The objective coupled cylindrical lens remains an interesting choice but there is little reasoning for it as described. For example, a galvo mirror could be used to scan a light sheet in synchrony with the detection optics (and at much higher speeds if necessary). This could either be achieved with a static cylindrical lens or an objective lens small enough to be located at 90 degrees, while allowing the detection objective to be mechanically refocused. In turn, this would also allow multi-color illumination with light sheets focused to the same depth and/or light sheet pivoting (using additional scanning optics) as desired. Since the authors report striping as a major concern, the method seems to overlook what could be done to easily address these artifacts, which are likely to be a greater source of difficulty than the beating heart. In this regard, the authors should cite Huisken et al., *Optics Letters*, 2007. Using the galvo method may even alleviate the difficulties of fast scanning owing to reduced mass and components providing drag. It would be good if the authors could explain the benefits of their approach thoroughly.

We thank the reviewer for pointing out that our original submission did not adequately communicate the broader implications of our work outside of OCPI microscopy. Both of the key advances demonstrated (fast mechanical scanning and parallelization across cameras) can be applied to accelerate other microscopy techniques. As the reviewer mentions, we could easily apply our approach to enhance the speed of a digitally scanned light sheet microscope (DSLIM), and this implementation would have some advantages—and disadvantages-- relative to our OCPI implementation. We have altered the manuscript to frame our contributions more broadly, and we have also clarified the relative advantages of OCPI: 1) OCPI obviates the need to synchronize scan and detection optics (which can be challenging at high speeds), 2) Our miniature illumination optics allow imaging of large samples (such as the cortex of a living, behaving mouse) that simply wouldn't fit if the illumination optics were larger. 3) Unlike DSLIM, the maximum axial extent of the volume imaged with OCPI is not limited by the field-of-view of the illumination lens. Rather it is set by the scan range of the mechanical positioner. Another minor point is that our system performs well with multi-color illumination because we form the light sheet with an achromatic lens. Finally, we have added a discussion of stripe artifact prevention, citing Huisken et al. The cited mSPIM technique sets a high standard for hardware-based stripe prevention, but we also cite

more recent work that has made strides in this area by using static diffusers that are more easily incorporated into our miniaturized illumination path: Taylor et al 2018 and Salili et al 2018.

5) The authors state “To our knowledge the fastest volume imaging demonstrated with OCPI or SPIM was 4 Hz, and this was with only a 32 μm scan range” Lemon et al. Nature Communications 2015, achieve 5 Hz imaging of the entire drosophila central nervous system and with the benefits of multi-view imaging for improved optical coverage. This was using DSLM but is equally applicable to SPIM (the speed is not determined by the illumination - in fact DSLM is substantially more challenging in this regard.)

This is a clear oversight on our part; we are grateful that it was caught and have modified the manuscript to reference Lemon et al 2015 as a high-speed imaging standard.

6) The authors state: “While “oblique” imaging methods (those that create the excitation sheet with the imaging objective 8,22,26,28) avoid these aberrations”. However, oblique methods do not avoid aberrations, there is substantial coma and a reduction of axial resolution owing to the non-orthogonal illumination-detection PSF overlap as shown in the supplementary information of Bouchard et al. Nature Photonics, 2015.

We agree with the reviewer that we were inaccurate in our statement that “oblique” imaging methods categorically avoid remote focusing aberrations. In our view this requires more subtle handling: oblique methods *can* avoid aberrations if they adhere to the criteria outlined by Botcherby et al 2007; Oblique Plane Microscopy (Dunsby 2008) is a pioneering example of this. The SCAPE method described in Bouchard et al, 2015 fails to meet these design criteria and indeed suffers from serious aberrations. We are aware that the SCAPE group has addressed this issue in a newer iteration of their microscope and that a similar design was published by the Kozorovitskiy lab (Kumar et al 2018).

7) The authors state: “Maximum readout rates of the top scientific CMOS camera manufacturers are all similar (Figure 1e), and these rates have not improved within 6 years, suggesting it may not be fruitful to wait for faster cameras.” This is the case for sCMOS cameras. However, Mickoleit et al., Nature Methods, 2014 used an ultrafast camera to image at nearly 5000 fps and 60 vol/s accordingly. This should be noted.

We have revised the manuscript to cite Mickoleit et al. 2014 and added a brief discussion of the tradeoffs between scientific CMOS cameras and general-purpose “ultrafast” cameras such as the PCO.dimax used in that study. The faster cameras are viable options in some scenarios, and in our revision we have taken care to avoid any suggestion otherwise. However we find that they are suitable for a narrower range of scenarios than the scientific models such as the PCO.Edge. This is because they exhibit much more noise and less photon efficiency than the scientific models. For instance, the current PCO.dimax exhibits 23 electrons rms noise and 50% peak quantum efficiency, while the PCO.Edge 4.2 shows 1.4 electrons rms noise and 82% peak QE. These numbers indicate that the PCO.dimax camera requires approximately 26x more light to achieve the same signal-to-noise ratio as the PCO.Edge 4.2. Thus the dimax requires more intense illumination and is best suited to very brief recordings in which rapid photodamage and bleaching are tolerable. We argue again (as in our response to point #1 above) that light efficiency remains an important consideration, even for light sheet microscopy methods. Another drawback is that current dimax cameras cannot sustain maximum framerate for very long; they acquire in brief bursts because the onboard camera memory fills faster than it can be emptied. For decades there have been a few cameras capable of extreme speeds for short bursts, but the sustained speed of cameras has been limited by the data bus, and after the initial sCMOS revolution this has been surprisingly slow to change.

8) The authors state “When compared to the 1.25 ms interval that would be possible with a true triangle wave it is apparent that the camera spent 37% of the time idle” this suggests that with 2 cameras the speed is 126% of a single camera operating at its full rate when combined with a faster focusing method. This detracts from the argument of parallelizing cameras before really solving the problem of fast focusing. Nevertheless, the authors have gone to some lengths to report where the method could be improved and the painstaking steps they have gone to to ensure that the system works as well as it does.

We agree that decreasing this idle time (and thus utilizing more of the camera throughput) would benefit the system greatly, and we are confident that this can be done if the problem is given sufficient engineering attention. However even without improving this parameter the system is highly scaleable when the scan improvements are taken together with the camera parallelization (4 cameras would achieve 252% of the speed of a single camera, 8 cameras would achieve 504%, etc). Rather than optimize the scan efficiency further our intention with this publication is to present the novel principles and leave this optimization for future work.

9) In general, regarding the assertion that artifacts resulted from the fish beating heart with a primary frequency at 15 Hz. It is unfortunate that this is only illustrated through lower frequency harmonics. Could the authors also illustrate this directly by imaging a single plane very fast to capture the 15 Hz with necessary Nyquist sampling (i.e. >30 Hz)? Also, it is simple to stop the heartbeat with excess anesthetic. This will change the neural activity for sure but then the loss of the oscillation can be shown in the power spectra.

We did not state that the primary frequency of the heartbeat was 15Hz. Rather we stated that 15Hz is the minimum sampling rate required to capture the largest harmonic that we observe in the power spectrum (7.5 Hz). The fundamental frequency in that recording is 2.5Hz, a factor of 3 less than 7.5Hz. We clarify this in the revised text. In addition to this clarification in our revision we chose to address two important points raised by the reviewer: 1) By sampling at only 20Hz we could not properly assess the strength of the artifact at higher harmonics, and 2) Without further evidence we cannot rule out that the “heartbeat” artifact has a non-biological source. We appreciate the reviewer’s suggestion of a specific experiment. We performed the experiment, imaging in a single plane at a higher rate to estimate power at frequencies up to 30Hz. We performed calcium imaging in the forebrain before and after administering a high dose of anesthetic (tricaine). Results are presented in a new supplemental figure (Fig. S6). We observed that putative heartbeat-related peaks in the spectrum disappear after stopping the heart with tricaine. We also analyze the power spectrum of the motion vectors found by our image registration algorithm and find corresponding peaks in the pre-tricaine dataset that are not present in the post-tricaine condition. This strongly suggests that the artifact is related to specimen motion rather than some other physiological correlate of the heartbeat. We also note that the fundamental heartbeat frequency differed in our new dataset (1.75Hz instead of 2.5Hz). This is not so surprising given the animal-to-animal variability present in the heart rhythm measurements of Luca et al 2014.

10) Another reason to image faster is to capture voltage dynamics rather than calcium as a proxy. The voltage spiking would be much faster and the signals dimmer, justifying the choice of LSFM as an imaging technology and the higher speeds accordingly. While I would not expect the authors to provide such data at this point, mentioning it as a future direction would strengthen the arguments.

We agree fully, as we also discussed in our response to point #1 above. We developed the microscope with voltage imaging in mind as a future application, but our original submission did not stress this enough. This is especially pertinent given very recent improvements in voltage indicator performance. Voltage indicators are approaching competitiveness with calcium indicators, and we are hopeful that they might soon replace calcium indicators as the standard tool for optical neurophysiology.

Response to feedback from Reviewer 2 (Our responses are in bold font)

- The authors discuss one of the important alternatives to mechanical scanning, i.e. remote focusing. They argue that this approach suffers from two major drawbacks: a partial loss of photons, and a degradation of the axial resolution. For the latter, I do not understand the line of reasoning as detailed in the supplementary materials: I feel that the calculation is flawed. It assumes that the intermediate optical system does not have a variable focal length, such as to successively bring the different z-planes in focus, as is precisely the principle of remote focusing. Without such possibility, the image formed on the sensor is obviously blurred as soon as the point source is away from the focal plane.

We are grateful to the reviewer for this comment, which has revealed that the argument made in the supplementary math document does not sufficiently describe the problem and our rationale. Perhaps the primary source of confusion is that our diagram and discussion only show a single lens, but variable focal length microscopes are implemented with two lenses: an objective lens (fixed focal length) and a tube lens (variable focal length). Since the objective lens focal length cannot be tuned, only a single plane (its native focal plane) can be imaged without aberration. This is true regardless of the focal length of the tube lens used, a fact established by the Abbe sine condition. A tuneable tube lens can correct for *defocus* aberration in planes outside of the native focal plane, thus far they do not correct for higher order aberrations (spherical, astigmatism, coma, etc) in an objective-specific manner. Our analysis accounts for this (assumes defocus aberration has been corrected), and we quantify only the severity of what is essentially spherical aberration. Our revised text and diagrams explain this in more detail, beginning with the objective+tube lens combination rather than immediately abstracting the system to a single lens.

In order to achieve aberration-free images of non-native object planes two approaches have been developed: 1) correcting aberrations with adaptive optics, and 2) setting the system magnification to meet the requirement for perfect volume imaging as described in Botcherby et al 2008 and our supplemental material (typically this means $M = 1.33$). The latter approach can also be viewed as aberration correction: aberrations induced by the sample-facing objective lens are reversed by the second lens (or set of lenses), and the 3D fluorescence distribution in the sample is copied to another location where it can be imaged again with higher magnification.

In our revised manuscript we have made substantial additions to the supplement detailing the above rationale.

- Current state-of-the-art volume scanning rate should be indicated more clearly. The limitation is not in the scanning rate per se, but in the scanning speed (there is always a trade off between axial range and scanning rate). The authors should indicate what is the maximum speed in scanning that is currently attainable, and how it compares with the scanning speed that they do reach with their approach?

We agree that scanning speed is more fundamental than the number of volumes imaged per second. For this reason we provided scan ranges along with our volume-per-second numbers in the original manuscript. The reviewer's suggestion to instead present results as speeds (i.e. units of mm/s) is a good one. In our revision we have added this quantity to our descriptions of the state-of-the art. We also chose to keep the volume/s numbers because we think they provide useful reference points for many readers, and also because the scan range affects the nature of the problem to some extent. For example, many commonly-used piezoelectric scanners simply cannot achieve the scan range that we demonstrate.

- The authors define the volume imaging rate as twice the scanning rate, because they image each section twice during a complete scan (bidirectional imaging). I think this is misleading and that the authors should use a conservative definition of the imaging rate as the inverse of the largest time interval between successive images of any given slice. As I understand it, their imaging approach leads to uneven time intervals between successive acquisitions. For the sections located at the very top or bottom of the volume, the actual imaging period thus

corresponds to the scanning period (and not the scanning half-period). The maximum volumetric imaging rate is thus 20Hz over 700um (which is already great !) but not 40Hz as claimed.

The reviewer's point is well-taken; with a bidirectional imaging paradigm the sampling interval depends on the axial location of each plane of the volume. As the reviewer states, in our "40Hz" bidirectional scanning demonstration the interval between successive samples at a depth near the edge of the scan range can be longer, approaching (but never reaching) 1/20 s (20Hz frequency). However we point out that the sampling interval actually alternates between this slower value and a much faster value. For the edge planes of the stack this faster value is on the order of the exposure time of the camera (somewhat longer because we don't use a true triangle wave) and thus shows an effect inverse to the one that the reviewer has highlighted, corresponding to a sampling rate greater than 40Hz. The average of the longer and shorter sampling interval is 1/40 s, corresponding to the 40Hz cited. We agree with the reviewer that an unqualified statement of 40Hz is misleading, but we find an unqualified statement of 20Hz to be misleading as well. The most accurate phrasing seems to be that the scan system supports an average sampling rate of 40Hz. In the revised manuscript we now present the rate as an average number. We leave intact our discussion of the advantage of frame interpolation to compensate for this non-uniform sampling rate, and more generally to compensate for the non-uniform sampling rate of any volume imaging technique that relies on axial scanning.

- When applied on zebrafish, the authors should indicate the total volume, inter-layer distance, and resolution for the large volume recordings at 10Hz (as is done for the smaller region ("In addition we imaged the whole brain of the fish at a rate of 10 Hz »))

We thank the reader for noting this omission; in the revised manuscript we state these numbers (inter-layer distance and resolution were equal to the recording of the smaller region).

Minor points :

- In the discussion, the authors write : « However, OCPI was historically unable to achieve the volumetric imaging speeds enjoyed by several more recent contributions ». Could they provide references and details of these recent contributions and how their performances compare with the present technique.

We have added specific imaging speed numbers to the discussion at the location noted, and we find that this strengthens the manuscript considerably. As in our original submission we also make image quality comparisons with these same techniques in the subsequent paragraphs.

- They also indicate that « SPIM, perhaps the closest relative to OCPI microscopy, shares these optical advantages and differs only in that the sample is translated rather than the optics ». I think that SPIM does not necessarily involve the motion of the sample. Vertical scanning is in fact generally performed by displacing the laser-sheet with a galvo. So I think this remark should be removed.

The originally published description of SPIM (Huisken et al 2004) required motion of the sample. Since then many alternative implementations have been developed, and some of these avoid translating the sample. To our knowledge these newer techniques have adopted different acronyms. For instance, the first demonstration of a galvo-scanned light sheet appeared as Digitally-Scanned Light Sheet Microscopy (DSLM, Keller and Stelzer 2008). However this comment highlights that in colloquial usage the term SPIM is associated with a broad family of techniques similar to the original SPIM method. In our revision we have taken care to qualify our statements about SPIM accordingly. The statement above now reads "The original SPIM implementation (Huisken et al 2004), perhaps the closest relative...."

REVIEWERS' COMMENTS:

Reviewer #1 (Remarks to the Author):

The authors have addressed all of my concerns. Just two minor points:

With regard to the discussion of DSLM. The passage is slightly confusing, the key feature of DSLM is scanning of the light sheet in the detection plane to produce a virtual light sheet. While it may also be that DSLM has often been used to scan the light sheet along the detection axis (to achieve volumetric imaging). I would recommend avoiding referring to this kind of light sheet repositioning as DSLM to avoid such confusion.

On page 18, the authors state that the highest efficiency reported in an oblique system is 21%. This should now be amended to reflect the recent Yang et al. Nature Methods, 2019, reporting the eSPIM system, which achieves far greater efficiency.

Reviewer #2 (Remarks to the Author):

The authors have made thorough modifications to their manuscript, and correctly addressed all the referee's remarks. I believe the current version deserves rapid publication in Nature Communications.

Second response to Reviewer #1 (Our responses are in bold font)

The authors have addressed all of my concerns. Just two minor points:

With regard to the discussion of DSLM. The passage is slightly confusing, the key feature of DSLM is scanning of the light sheet in the detection plane to produce a virtual light sheet. While it may also be that DSLM has often been used to scan the light sheet along the detection axis (to achieve volumetric imaging). I would recommend avoiding referring to this kind of light sheet repositioning as DSLM to avoid such confusion.

We agree that this was confusing as written. We have revised the section so that DSLM is not confused with this kind of scanning, though we still cite the original DSLM manuscript as an example of such scanning.

On page 18, the authors state that the highest efficiency reported in an oblique system is 21%. This should now be amended to reflect the recent Yang et al. Nature Methods, 2019, reporting the eSPIM system, which achieves far greater efficiency.

We've updated our discussion to include this very recent contribution. Indeed the authors demonstrate impressive efficiency. We've also discussed briefly that their approach still suffers from aberrations, and therefore diffraction-limited imaging can only be achieved in a limited volume without scanning the sample stage. Quoting from Yang et al 2019:

“Within an imaging volume of about 70 μm in y , 20 μm in z and Galvo scan range of ~ 100 μm in x , the aberration was minor and our microscope had a performance close to the diffraction limit with a Strehl ratio of 0.90 (Supplementary Figs. 7 and 8). This imaging volume is similar to that reported by lattice light-sheet microscopy⁶ and can be further increased by scanning of the sample stage if necessary.”